# LLM Chess: Benchmarking Reasoning and Instruction-Following in LLMs through Chess

## Abstract

We introduce LLM Chess, an evaluation framework designed to probe the generalization of reasoning and instruction-following abilities in large language models (LLMs) through extended agentic interaction in the domain of chess. We rank over 50 open and closed source models by playing against a random opponent using a range of behavioral metrics, including win and loss rates, move quality, move legality, hallucinated actions, and game duration. For a subset of top reasoning models, we derive an Elo estimate by playing against a chess engine with variably configured skill, which allows for comparisons between models in an easily understandable way. Despite the simplicity of the instruction-following task and the weakness of the opponent, many state-of-the-art models struggle to complete games or achieve consistent wins. Similar to other benchmarks on complex reasoning tasks, our experiments reveal a clear separation between reasoning and non-reasoning models. However, unlike existing static benchmarks, the stochastic and dynamic nature of LLM Chess uniquely reduces overfitting and memorization while preventing benchmark saturation, proving difficult even for top reasoning models. To support future work on evaluating reasoning and instruction-following in LLMs, we release our experimental framework, a public leaderboard, and a dataset of associated games.[1]

## 1 Introduction

Chess has long been viewed as an application for artificial intelligence (AI) since its inception, often being one of the first domains in which new technologies are used (Prost, 2012). The idea of computer chess was pursued by the founders of AI, who viewed it as an exciting application in which advances could spur developments in other fields (Turing, 1988; Wiener, 2019; Shannon, 1950). In fact, chess is often referred to as the 'drosophilia of AI', in that it both is a worthy testbed for experiments and also has guided the field's development (Simon & Schaeffer, 1992; McCarthy, 1990; Ensmenger, 2012). As such, chess also has often been used to study cognitive abilities and decision making in humans (Groot, 1978; Simon & Chase, 1988; Sala et al., 2017; Sala & Gobet, 2017; Burgoyne et al., 2016; Blanch, 2022; Rosholm et al., 2017; Jankovic & Novak, 2019).

Since the 1950s, chess engines have been created with the hopes of beating humans, achieving various levels of success along the way. As time progressed, these engines advanced both through hardware and algorithmically, until reaching their current most powerful form with neural networks (Bernstein & de V. Roberts, 1958; Adel'son-Vel'skii et al., 1970; Newborn, 1979; Condon & Thompson, 1983; Campbell et al., 2002; Newborn, 2012; Silver et al., 2017). While certain architectures and algorithms applied to chess have seen success elsewhere, these chess engines are explicitly tailored to chess games, unable to generalize.

Recently, large language models (LLMs) have shown incredibly competent performance in many diverse fields (Brown et al., 2020; Touvron et al., 2023; Thirunavukarasu et al., 2023; Liu et al., 2023; Wu et al., 2023b; Wei et al., 2022; OpenAI et al., 2024; DeepSeek-AI et al., 2025), leading many to wonder whether they may play an important role in achieving artificial general intelligence (Bubeck et al., 2023; Feng et al., 2024; Mumuni & Mumuni, 2025). Additionally, tools like reinforcement learning and test-time scaling approaches have been shown to greatly increase reasoning abilities,

---

[1] Our code is available at `https://anonymous.4open.science/r/llm_chess_anon-5CCE`

accelerating the promise of a general reasoner (Chen et al., 2024; Shao et al., 2024; DeepSeek-AI et al., 2025). While chess engines can now regularly beat humans, the game has not yet sufficiently been tested on LLMs, which ideally would possess such general characteristics that they could excel at any complex reasoning task, whether it be math, coding, or gameplaying like chess. As we start to design models with more general capabilities, what is old becomes new again: the large combinatorial spaces, long-horizon planning, and dynamic nature of chess all present thorough challenges for LLMs. Continuing the tradition of using chess to test and gain insights into current model capabilities, we present two main contributions:

1. We introduce LLM CHESS, a benchmark assessing both reasoning and instruction-following in the context of chess. Central to our benchmark is agentic interaction: by having LLMs play chess through autonomously selecting actions within a conversation, the difficulty comes not only in reasoning about the board and choosing the best move, but also how to formulate these choices. Unlike other reasoning benchmarks that can be contaminated or easily saturated, LLM CHESS is extensible by scaling the difficulty of the opponents and is not reliant on static board positions that can be included in training data.

2. We evaluate over 50 models on LLM CHESS, showing that the domain of chess continues to present a challenging and informative reasoning task when applied to LLMs. We find that currently only the most powerful reasoning-enhanced LLMs can consistently beat a random agent, even when we let them query for legal moves. When playing against engines, these powerful models still fare poorly, with o3 (low) only achieving a 758 Elo in LLM CHESS. Through extensive ablations on specific parts of the game, we find that LLM performance varies widely based on the format of the conversations and prompt, suggesting a lack of robustness in their reasoning abilities.

Altogether, our comprehensive experiments show that chess is a worthy testbed for benchmarking the reasoning and instruction-following ability of LLMs and that current state-of-the-art models lack the ability to generalize their strong reasoning performance to be as impressive in chess as in other domains.

## 2 LLM CHESS

Here we introduce LLM CHESS (Figure 1), explaining our design choices and the metrics we use to score the models.

### 2.1 DESIGN

In chess, an action taken by one side is referred to as a half-move or ply while two concurrent plys are referred to as a move, one by white, the other by black.[2] At each ply, we initiate a conversation with the end goal of outputting a valid chess move. We format all moves in Universal Chess Interface (UCI) format, a commonly used notation for chess engines (Huber & Meyer-Kahlen, 2000). Each conversation consists of several turns, where each turn an LLM is prompted with instructions to output a valid action. We offer three actions to the LLM: 1) `get_current_board`, which fetches and presents the state of the current board using a unicode board, 2) `get_legal_moves`, which fetches a list of legal moves in UCI format, and 3) `make_move`, which takes a UCI-formatted string as input, adjusts the board state with that move, then ends the LLM's turn.

We provided the opportunity to retrieve full board states and legal moves through tool calls while excluding move history, creating an agentic approach that balances realism with practical testing needs. Full design justifications are in Appendix A. Ablations on these choices are presented in Section 3.4. We implement our LLM in an agentic setting using the AG2 framework (Wu et al., 2023a; Wang et al., 2025).

We cap each game at 100 moves (200 plys), have a max of 10 conversation turns per ply, and allow a max of 3 attempts per conversation turn for the LLM to provide a legal action or move. The LLMs view each ply as independent of all others, as we do not provide any game history. While this differs

---

[2]When it is clear that we are only discussing one side's actions, we occasionally overload move to refer to a ply, i.e., making a move in a ply refers to a single piece movement for that specific ply.

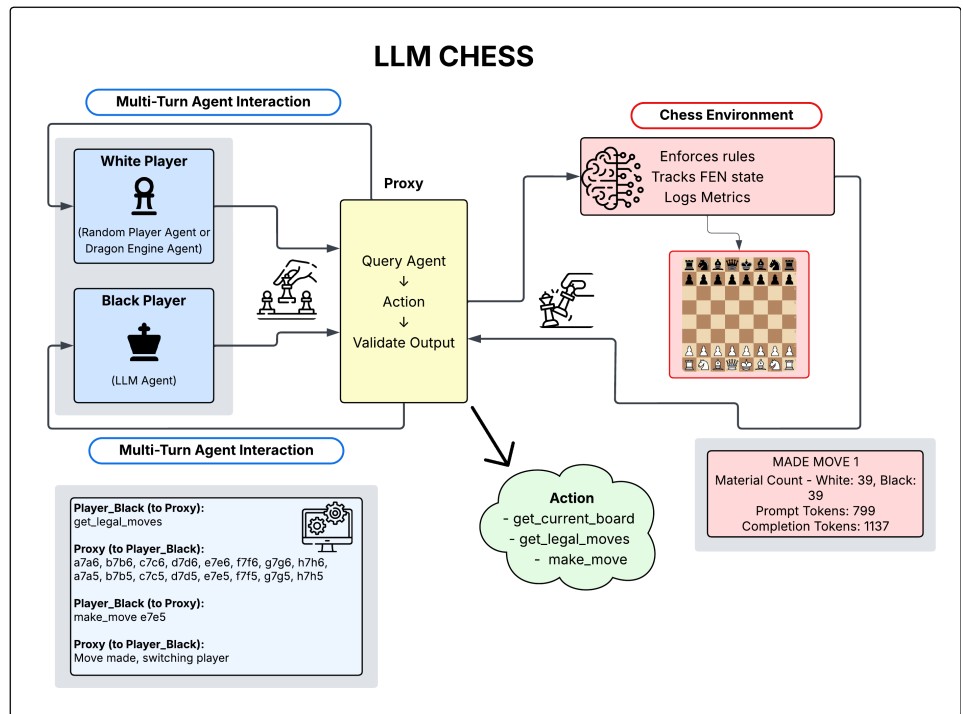

Figure 1: Overview of the LLM CHESS benchmark. White and Black player agents (random or engine for White, LLM for Black) interact with a central proxy that issues agent queries, validates outputs, and invokes one of three actions (`get_current_board`, `get_legal_moves`, `make_move`) The Chess Environment enforces the rules, updates and logs the FEN state, and records per-move metrics for downstream analysis.

from humans who know their previous moves when playing chess, this aligns more with the machine setting where a model should be able to make the best move given the board state alone. Importantly, this setting does not eliminate the need for long-term planning: models must continue to be aware of how the moves they choose will impact future board states.

Instructions provided to the LLM to initiate the conversation and resulting from various actions are presented in Appendix D. From preliminary testing, we somewhat surprisingly found many LLMs performed poorly against random agents. So, we split our evaluation into two phases: first, we evaluate a wide set of models against random agents to get a general sense of their abilities. Second, on particularly good models, we play them against a chess engine with variably configured skill.

**Random Agent** We benchmark over 50 models by playing 30 games as black against a random agent, defined as a player who always chooses a move at random from all legal moves. We choose a random agent first because we want to focus on practical game-playing ability while removing skill as a main focus, i.e., to see if the model can play and finish a game of chess in the simplest setting. The goal of this phase is to isolate instruction-following abilities while minimizing opponent difficulty. While we only play 30 games, we note that our end-goal is not to precisely rank models based on their performance vs random, but instead to see whether these models can both 1) exhibit sufficient performance against a random agent to be worth playing against a powerful chess engine, and 2) do not exhibit simple instruction-following errors that cause incomplete games. In this sense, the random play phase can be seen as a cost-effective gating mechanism for reasoning evaluation with a chess engine, albeit one that can still tell us a lot about the models.

**Chess Engine** From the initial models, we choose a subset of promising models to play against Komodo's Dragon 1 engine, which can be set at various skill levels from 1-25. As an estimate, skill 1 is around Elo 250, then each subsequent skill level is a 125 boost in Elo based on chess.com games (Kaufman & Lefler, 2020). Since chess.com is one of the most popular online chess platforms,

having over 200 million members (Chess.com, 2025a), this lets us ground our LLM performance in the real world. We run experiments against Dragon 1 at 30 games per skill level and depending on the model, run experiments for a variety of skills, starting at skill 1 and getting as high as skill 10, representing Elo scores of 250 to 1375 on chess.com. While currently we do not evaluate with too high of skills, our framework permits easy extensibility: as LLMs become better and better, we can increase the difficulty of the opponents to prevent saturation. The goal of this phase is to evaluate reasoning abilities in our simple agentic setting on models we know can perform well without instruction-following errors. This should mimic real-world agentic settings in which we need models to have some minimal instruction-following abilities before they can successfully solve a task.

## 2.2 METRICS

LLM CHESS evaluates LLMs by playing full chess games. However, we also evaluate the reasoning ability of the LLM with various per-ply metrics rating the quality of each move, as well as the instruction-following ability by examining how the model engages with our agentic structure.

**Per-model**  The main way we quantify performance is to calculate a LLM's Win/Loss percentage against an opponent, which is the difference between wins and losses as a percentage of total games:

$$\text{Win/Loss} = \frac{1}{2}\left(\frac{\text{llm\_wins} - \text{opponent\_wins}}{\text{total\_games}}\right) + 0.5$$

Win/Loss admits easy interpretability: 50% means a model has equal wins and losses. To win a game, LLM must checkmate its opponent. LLMs can lose or draw in the following ways: 1) Chess-based. The LLM could lose through checkmate by the opponent or draw due to various rules (stalemate, insufficient material, seventy-five moves without a capture or pawn move, fivefold repetition, or the game reached 100 moves). 2) Instruction-based. The LLM loses if it reaches the maximum number of conversation turns without making a move (10) or if it reached the maximum number of attempts (3) at a conversation turn without selecting a valid action. We call failures here instruction-following errors. 3) Model errors. These are errors due to the model or how it's served like timeout for reasoning models. We exclude all games with these errors when playing against a random agent so we could better analyze behavior, but include them when playing against Dragon 1 to simulate what would happen in a real-world scenario.

While Win/Loss is helpful for observing the quality of LLM performance against weaker opponents, it is less grounded in the world of chess. So, for LLMs that perform sufficiently well against random agents and against the engine at various skill levels, we calculate Elo (Elo, 1978). Normally Elo ratings update dynamically between players, but here we treat each engine opponent's rating $R_i$ as fixed and encode the LLM's game outcomes as $S_i \in \{1, 0.5, 0\}$. Under Elo theory, the expected score $E_i(R)$ for a player with rating $R$ against opponent $i$ with rating $R_i$ is:

$$E_i(R) = \frac{1}{1 + 10^{(R_i - R)/400}}.$$

Rather than updating $R$ incrementally, we find the maximum-likelihood Elo rating $\hat{R}$ by solving $\sum_i \left(S_i - E_i(\hat{R})\right) = 0$. Around $\hat{R}$, the observed Fisher information $\mathcal{I}(\hat{R}) = \sum_i E_i(\hat{R})\left(1 - E_i(\hat{R})\right)(\ln 10/400)^2$ yields a standard error $\text{SE} = 1/\sqrt{\mathcal{I}}$ and thus a 95% confidence interval for the Elo rating $\hat{R} \pm 1.96\,\text{SE}$ (Glickman, 1999). We detail the exact skill levels we evaluate against for each model in the experiments section and the full Elo calculation algorithm in Appendix B.4.

**Per-game**  For each game, we calculate the number of moves per game and the reason for each loss. We also record other metrics focused on instruction-following throughout the game that do not depend on the quality of the moves. For `get_current_board` and `get_legal_moves` we calculate the average number of times that action was called per ply. We also calculate the average number of times `make_move` was called but resulted in an invalid move, as well as the average number of invalid actions that were selected.

**Per-ply**  Besides analyzing performance on a game level, we also calculate the performance per ply. After the LLM calls `make_move` in each ply, we calculate the Win% (Equation (1)), the chance of

winning a game from the given position as defined by Lichess (Lichess, 2025). This analysis is based on centipawns, which are calculated by Stockfish representing how much worse the player's move was than the engine's (Linville, 2023). We present the Win% for the LLM averaged over each ply, which tells us whether the LLM held a more favorable position throughout the game.

$$\text{Win\%} = 50 + 50 * (2/(1 + \exp(-0.00368208 * \text{centipawns})) - 1) \tag{1}$$

Then, based on the difference in Win%, $\Delta = \text{Win\%}_{\text{before move}} - \text{Win\%}_{\text{after move}}$ (where a higher $\Delta$ means the player's Win% decreased), we can calculate Blunders, Mistakes, and Inaccuracies, common classifications of moves used by online chess platforms, following the Lichess cutoffs (Lichess, 2023):

$$\text{Judgment} = \begin{cases} \text{Blunder} & \text{if } \Delta \geq 30 \\ \text{Mistake} & \text{if } \Delta \geq 20 \\ \text{Inaccuracy} & \text{if } \Delta \geq 10 \end{cases} \tag{2}$$

We present the average Blunder, Mistake, and Inaccuracy rate per ply, as well as Best, the rate in which the LLM selected the best move as identified by Stockfish. We note that since our Win% scores are based on centipawns, these metrics can depend on the hyperparameters of Stockfish. Additional details for centipawn calculations are available in Appendix B.1.

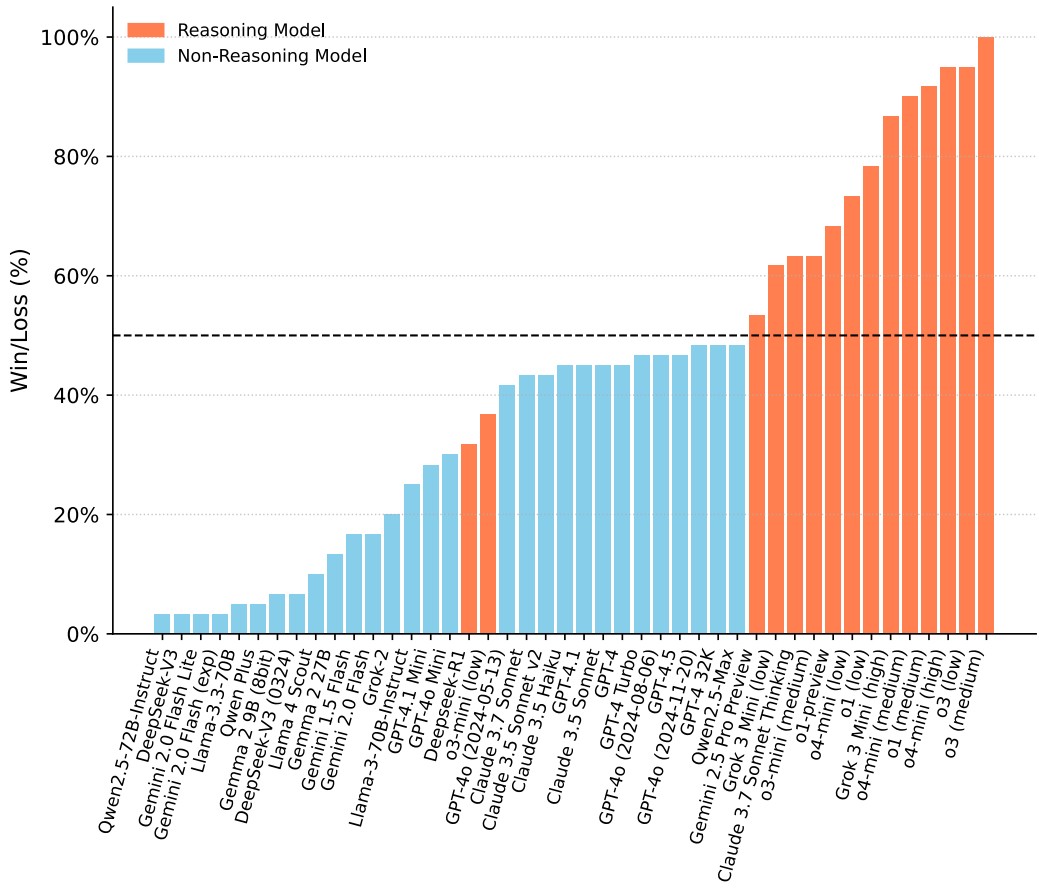

Figure 2: Win/Loss of LLM players versus random opponents. The dashed line marks a Win/Loss of 50%, which represents an equal amount of wins and losses.

## 3 EXPERIMENTS

By default, all LLMs are run with a temperature of 0.3 and a Top P of 1.0, with some exceptions. We run both regular LLMs as well as those trained for enhanced reasoning capabilities, which we term

reasoning-enhanced LLMs. Such LLMs have a separate space for thinking (e.g., a dedicated thinking tag in their chat template before the assistant response) indicating special training for reasoning, similar to o1 (OpenAI, 2024) or DeepSeek-R1 (DeepSeek-AI et al., 2025). More details about the models we evaluate on and how they are run is detailed in Appendix B.

## 3.1 LLMs vs. Random

We present the Win/Loss of 44 LLMs versus a random agent for 30 games in Figure 2. Most notably, we find that most models are not able to consistently beat a random agent; in fact, only models with reasoning abilities are able to perform better than 50%. To analyze the reasons behind this poor performance, we present per-game metrics including how the LLMs won and lost in Table 1. Note that the only way the LLM (black) can win is through a checkmate. For each of these metrics, we present the average over all reasoning and non-reasoning models, as well as on the two top and bottom reasoning and non-reasoning models.

Table 1: Per-game metrics for Reasoning (shaded) vs Non-Reasoning models. We choose the top and bottom two models in each category (ranked among 15 reasoning, 29 non-reasoning models) based on Win/Loss from among all models with a Win/Loss over zero. We include the percent of losses due to errors in instruction-following (Instruction) or checkmates by white (MateW), as well as the amount of draws (Draw), checkmates by black (MateB), and average moves over all games.

| Model | Instruction (%) | Draw (%) | MateW (%) | MateB (%) | Avg Moves |
|---|---|---|---|---|---|
| Reasoning Avg | 24.4 | 30.2 | 0.0 | 45.4 | 93.7 |
| Non-Reasoning Avg | 71.9 | 24.6 | 2.8 | 0.7 | 73.9 |
| o3 (medium)[1] | 0.0 | 0.0 | 0.0 | **100.0** | 40.1 |
| o3 (low)[2] | 0.0 | 10.0 | 0.0 | 90.0 | 63.5 |
| Qwen2.5-Max[1] | 0.0 | **96.7** | 3.3 | 0.0 | **197.4** |
| GPT-4o (2024-11-20)[2] | 0.0 | 90.0 | **6.7** | 3.3 | 194.9 |
| o3-mini (low)[14] | 36.7 | 53.3 | 0.0 | 10.0 | 139.3 |
| Deepseek-R1[15] | 60.0 | 16.7 | 0.0 | 23.3 | 88.2 |
| Gemini 2.0 Flash Lite[28] | **90.0** | 0.0 | **6.7** | 3.3 | 90.3 |
| Qwen2.5-72B-Instruct[29] | **90.0** | 6.7 | 3.3 | 0.0 | 64.1 |

Our results indicate that reasoning LLMs dramatically outperform non-reasoning models in our random-opponent setting. Reasoning models have an average win rate of 45.4% with the top performers achieving close to 100%, whereas non-reasoning models have an average win rate of 0.7% with one of the top performers achieving only 3.3%. This performance gap is further supported by a three-fold reduction in instruction-following errors: 71.9% for non-reasoning models vs 24.4% for reasoning models. Lastly, non-reasoning models almost always draw if they don't have instruction-following issues. Interestingly, these models have a similar percentage of draws compared to reasoning models (24.6% vs 30.2%). While these statistics demonstrate that enhanced reasoning capabilities substantially improve both instruction-following and overall game performance, only one LLM was able to win every game against a random agent, indicating poor real world performance.

Table 2: Per Ply Classification Rates (%) for Reasoning (shaded) vs Non-Reasoning Models.

| Model | Blunder ($\downarrow$) | Mistake ($\downarrow$) | Inaccuracy ($\downarrow$) | Best ($\uparrow$) |
|---|---|---|---|---|
| GPT-4.1-mini | 31.3 | 8.7 | 13.4 | 4.1 |
| o4-mini (low) | 11.2 | 3.5 | 5.5 | 10.8 |
| o4-mini (medium) | **4.2** | **1.1** | **4.0** | **19.5** |

To see how models perform throughout the game, we present per-ply metrics on a handful of models performing at various levels in Table 2. Our results show that the provided reasoning models make far fewer bad moves and substantially more "Best" moves than GPT-4.1-mini, the representative non-reasoning model. For example, o4-mini (medium) blunders only 4.2% and mistakes 1.1% of the time per ply, compared to 31.3% blunders and 8.7% mistakes for GPT-4.1-mini. Furthermore, o4-mini (medium) selects the "Best" move 19.5% of the time versus just 4.1% for GPT-4.1-mini.

These results confirm that enhanced reasoning capacity reduces catastrophic errors while boosting tactical decision making.

Notably, we also ran experiments on over 10 models that have a 0% Win/Loss, often resulting from difficulties with instruction-following. We present these models in Table 6. We also present additional results for some models on more games in Appendix C.

## 3.2 LLMs vs. Chess Engine

While random agents are a good test of LLMs' abilities to complete games, they often make moves that are nonsensical and are not realistic as a chess opponent. As such, some LLMs are able to perform very well against random agents: the best models o3 (medium/low) and o4-mini (high) have a Win/Loss of at least 90%. To increase the difficulty of the games and ground LLMs in real-world performance, we now focus on the most powerful models (i.e., a subset of reasoning models) to play against Dragon 1: o3 (low), Grok 3 Mini (high), o4-mini, o3-mini.

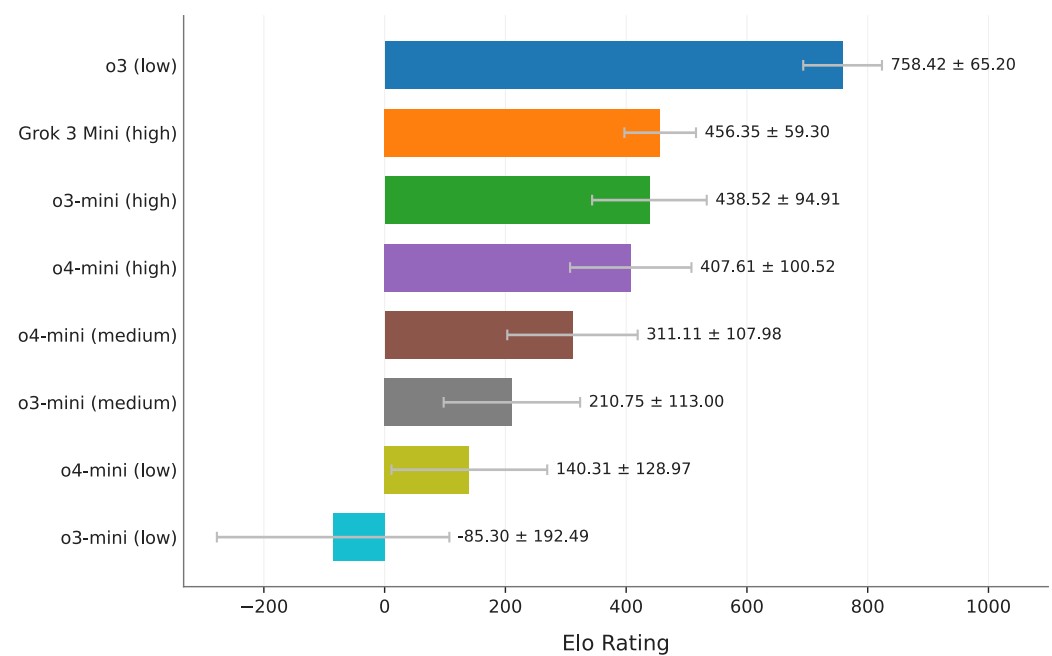

Figure 3: Elo of top reasoning models estimated using Dragon 1.

Figure 3 reports estimated Elo ratings (±95 % CI) for o3 (low), Grok 3 Mini (high), o4-mini, o3-mini when playing at least 30 games against Dragon 1 at skill 1. For o3 (low) and Grok 3 Mini (high) we play against all skills 1–5 (Elos 250–750), with the former additionally playing against skill 10. These Elo estimates confirm several key insights. First, better reasoning models have higher strengths in LLM Chess. For example, the o4-mini models dominate the o3-mini models on all but high reasoning effort, where they perform similarly. Second, even the strongest LLM in our study, o3 (low), peaks at an adjusted Elo of about 758, ==which remains only slightly above the average and far below the human master level, underscoring how far LLMs lag behind specialized chess engines and general human gameplay.== We include more about the models, skills each played against, real-world comparisons, and Elo calculations in Appendix B.

## 3.3 Exploring Test-time Scaling

**Scaling Deep** We show o1, o3, o4-mini, and Grok 3 Mini at various reasoning levels vs a random agent in Figure 4a. Similar to other reasoning domains, we find scaling with more tokens improves performance on LLM Chess, with increases of up to 15% from low to medium, and 20% from low

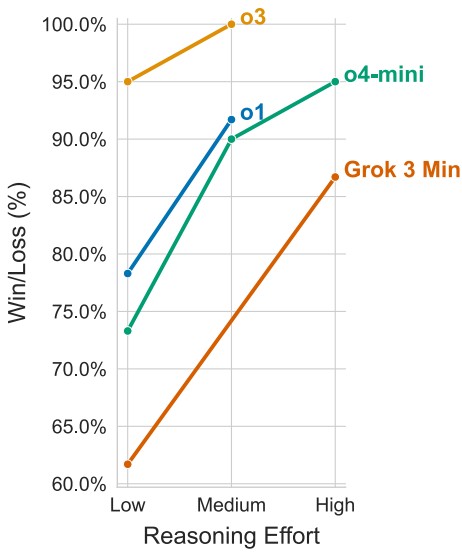 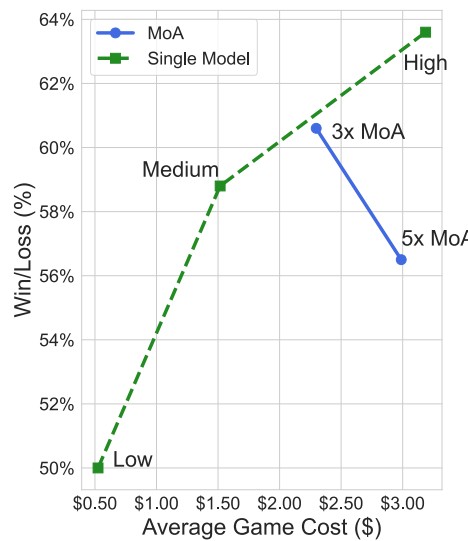

(a) Scaling deep with increased reasoning effort. (b) Scaling wide with o4-mini and MoA.

Figure 4: Performance comparisons of reasoning models. (a) Win/Loss when scaling with variable reasoning effort. (b) Cost-performance tradeoff for Win/Loss with o4-mini variants at each possible reasoning effort along with 3x and 5x MoA using o4-mini (low) as the proposer and o4-mini (medium) as the aggregator.

to high.[3] To directly compare performance on LLM Chess to performance on other domains, we calculate the correlation between scores on LLM Chess and LiveCodeBench (Jain et al., 2025), a difficult competitive coding benchmark, finding a moderate positive correlation. This signifies that LLM Chess uses reasoning abilities, though even the best models struggle to be as good as they are in other domains. See Appendix C.5 for further discussion.

**Scaling Wide** Besides increasing the number of tokens one model uses, we also run experiments using multiple instances of the same model in parallel. To do so, we apply a Mixture-of-Agents (MoA) approach where at each step of the conversation we have multiple proposer model calls fed into a separate aggregator model that provides the output (Wang et al., 2024). We run two settings on 30+ games with black against Dragon 1 skill 1 using either 3x and 5x o4-mini (low) as the proposers and always o4-mini (medium) as the aggregator. Results are in Figure 4b. Performance with 3x MoA reaches above o4-mini (medium), but 5x MoA performance is slightly lower. Though there are small differences between these approaches, in practice they all perform relatively similarly. This suggests that scaling the number of proposed moves doesn't yield significant improvements, unlike the benefits we see from scaling reasoning effort. Additional MoA experiments are in C.2, suggesting benefits can come from pairing models with poor instruction-following but strong reasoning capabilities with non-reasoning models that follow instructions well.

### 3.4 ABLATIONS

We design three types of ablations on o4-mini (low) and Grok 3 Mini (low) by varying the actions we present to the model during the conversation (Actions), the state of the board from the LLM's perspective (Board Representation), and adding or removing information the LLM has access to during the conversation (Changing Information). In each of the settings in each category we run 30 games per model against a random agent with the LLM playing as black (unless stated otherwise). Results are in Table 3, with more detail in Appendix C. With these results, we see performance varies widely, showing the lack of robustness in reasoning in the chess setting.

---

[3]Empirically, we notice that as we try to run OpenAI models with higher reasoning effort, they are more likely to result in a timeout. See Appendix E for further discussion.

Table 3: Win/Loss on ablations. Each is run with 30 games vs a random agent. LLM CHESS is the baseline.

| Setting | Grok 3 Mini (low) | o4-mini (low) |
|---|---|---|
| LLM CHESS | 61.7 | 73.3 |
| **Actions** | | |
| Always Board State | 66.7 | 83.3 |
| Always Legal Moves | 68.3 | 93.3 |
| Only `make_move` | 71.7 | **96.7** |
| **Board Representation** | | |
| ASCII | 63.3 | 88.3 |
| FEN | 63.3 | 95.0 |
| LLM as White | **78.3** | 83.3 |
| **Changing Information** | | |
| No Legal Moves | 36.7 | 86.7 |
| Previous Moves | 75.0 | 76.7 |
| Previous Moves + Only `make_move` | 66.7 | 95.0 |

Overall, we find that simplifying the agentic scenario by removing actions and instead supplying the removed information directly in the prompt without offering the associated tools shows an increase in performance on both Grok 3 Mini (low) and o4-mini (low). In both cases, offering only `make_move` offers substantial improvements in Win/Loss, with o4-mini (low)'s performance increasing by over 20%. This signifies the difficulty of reasoning models engaging in agentic interactions in LLM CHESS. Performance with both an ASCII board and FEN is similar to our default setting for Grok 3 Mini (low), while for o4-mini (low) we see performance improve by over 15% in both cases, reaching 95% for FEN. This suggests that some LLMs have similar performance across board representations, while some have trouble generalizing.

Though LLM CHESS's agentic setting can be challenging for some models, a major advantage given to the model is their ability to query for legal moves with `get_legal_moves`. When removing this ability, we see a decline in model capabilities of almost 30% for Grok 3 Mini (low), though we see an increase of 10% for o4-mini (low), meaning that some LLMs need help while others may be better off using their own internal reasoning processes. We also experiment with including the previous moves, finding that performance can increase but most often does not result in a substantial benefit.

## 4 RELATED WORK

**Chess and AI** Transformers have been applied to chess in both foundation and domain-specific settings. While prior work has suggested that large language models (LLMs) display surprising competence in chess (Dynomight, 2024; Acher, 2023), these findings often rely on a small set of models, static PGN completions, or idealized prompting conditions. Studies such as the Chess Transformer (Noever et al., 2020), Chessformer (Monroe & Chalmers, 2024), and BERT-based rule learners (DeLeo & Guven, 2022) demonstrate improved move legality and opening play, but confine game play to offline or single-turn evaluations. More recent work has involved fine-tuning transformer architectures directly on a large-scale chess corpus, such as ChessGPT (Feng et al., 2023) and Amortized Planning Transformers (Ruoss et al., 2024), with the latter treating chess as a planning problem. While these approaches show promise, they are typically assessed on win rate or move legality, focusing little on instruction-following or reasoning. For LLMs, several open-source efforts have attempted to evaluate on chess tasks, such as by creating frameworks letting humans play against LLMs (Carlini, 2024), having LLMs play against each other (Risdal, 2025), or having LLMs play against chess engines (Ndzomga, 2024). Other analyses examine how LLMs internalize chess rules from PGNs (Stöckl, 2021) and how LLMs can predict chess puzzle difficulty (Miłosz & Kapusta, 2024), or they include chess as part of a larger benchmark (Khan et al., 2025). While these frameworks provide initial insights, they typically focus only on outcome-level metrics such as win

rate or Elo, often over a narrow set of models in a basic setting. In contrast, our benchmark uses a diverse model pool in a simple agentic environment with a minimal set of tools, revealing fragility in instruction-following, real-time play, and strategic reasoning.

**Strategic Reasoning and Game Benchmarks**   Our work builds on a growing field of literature that poses games as testbed for strategic and multi-step reasoning. GTBench (Duan et al., 2024) and ZeroSumEval (Khan et al., 2025) leverage inter-model competition to assess strategy and robustness, while ChatArena (Wu et al., 2023c) and MastermindEval (Zhang et al., 2024) extend the space of game evaluation into multimodal and logic-heavy tasks. Additional studies in multi-game consistency (Toshniwal et al., 2022) highlight gaps in rule following and tactical depth when LLMs pivot between environments. While these efforts highlight the strengths and limitations of LLMs in planning, consistency, and rule/instruction following, they are typically spread across tasks or lack depth. Chess on the other hand, is a deeply studied environment with transparent rules, interpretable decision sequences, and established baselines. Our benchmark combines all of these strengths in a reproducible testbed that evaluates both instruction-following and reasoning.

## 5 CONCLUSION

Chess has long been an important factor in the development of AI systems. However, LLMs, today's most powerful generalist models, have not been sufficiently tested on the domain, missing out on the insights that have historically been made by doing so. To remedy this, we introduced LLM CHESS, a benchmarking framework for reasoning and instruction-following in LLMs in chess. Compared to standard reasoning benchmarks, our setting is more difficult: unlike math or coding where LLMs are reaching the level of seasoned experts, models evaluated by LLM CHESS are weak and many cannot consistently beat even a player making random moves. Importantly, as LLMs become better, LLM CHESS can still be used without fear of saturation. Built around a chess engine, it allows for extensibility through dynamic difficulty adjustment, as well as resistance to memorization thanks to the combinatorial richness of chess, offering a reasoning benchmark designed to remain informative as models improve.

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

## A    DESIGN CHOICE JUSTIFICATION

We acknowledge that our benchmark includes settings that deviate from what you would find in the real-world. However, our goal was not to perfectly mimic humans playing chess but instead to use chess as a testbed to evaluate different aspects of LLMs including instruction-following and measuring the abilities of reasoning models beyond simple move completion settings. The main deviation was our introduction of tools, i.e., the ability to see the current board or legal moves with a tool call. While it may seem unorthodox, the results show that introducing such an agentic approach is useful in measuring instruction-following, a central goal of the benchmark: of the 44 models we tested vs a random opponent with positive performance, we see instruction-following errors are responsible for 71.9% of all games ending for non-reasoning models and 24.4% for reasoning models, on average. Even more powerful models we might not expect to have such errors, e.g., Deepseek-R1 or o3-mini (low), show non-negligible problems with instruction following.

The main design choices we made beyond the agentic setting was supplying the current board and legal moves but not providing the previous moves. We justify our other choices below, which we will add to the limitations section of our paper:

**Board State**    We assume that the model is able to see the full board at any time, differing from some models that see only the previous moves or a pgn description of the game. We chose this to be more similar to what a human player or chess engine would see.

**Legal Moves**    We decided to provide legal moves to simplify the benchmark, as current capabilities of models are not yet enough to play consistently without providing the legal moves. See Table 6 in Appendix B, where not including legal moves causes a decrease in Win/Loss of 30% for grok-3-mini-low and 10% for o4-mini (low) compared to the baseline (note for legal moves and its comparison we use the FEN setting as without legal moves, we cannot know castling rights or en passant). Essentially, including the legal moves was a practical concern: it allows us to have more granularity between models by boosting their performance and preventing clustering at low performance due to move failures.

**Previous Moves**    We chose not to include previous moves to increase similarity with existing AI approaches for playing chess. Chess engines like Stockfish can evaluate the best move given a board state alone without any move history. If LLMs are to reach the level of other AI systems in chess, we believe it is helpful to see them perform under these same constraints. So, we decided not including previous moves would result in a more challenging and ideal goal for LLMs. This decision not to include previous moves was made during the initial trials of the benchmark during its creation, while experimenting with different prompts across a subset of models. During these experiments, including the history bloated the prompt and made some of the models struggle more with instruction-following, so we also chose this setting as a practical concern. Moreover, in our paper, we analyzed performance of including previous moves in our ablations in Table 6. We found that while including previous moves in the prompt did improve performance, the change was varied and altogether not drastic, suggesting that if anything, the previous moves can help reduce complexity and blunders, not increase them.

## B    EXPERIMENTAL SETTINGS

We ran all LLMs with a default temperature of 0.3 and Top P of 1.0 for the models that took them as parameters (some models like OpenAI's reasoning models don't take a temperature). If models like Deepseek-R1 have a recommended temperature (0.6), we try to use that instead. We define "reasoning-enhanced" models as those that are specifically advertised/characterized by their developers as "reasoning" (e.g. OpenAI) or "thinking" (e.g. Anthropic, Google) without going into detail as to how those models are built (generally RL and test-time compute are mentioned, yet the detail and disclosure varies). On the surface the reasoning enhanced models manifest their nature by splitting the response into two sections: 1) reasoning/thinking intermediary, delimited via a special section in the chat template (such as a think tag) or residing in a separate API response section (e.g. thinking block in Anthropic API), and 2) the final answer. E.g., aligned with their advertised functionalities,

we designate as reasoning the following: all "o" family of models (e.g., o1, o3, o4-mini), Claude 3.7 Thinking, Grok 3 Mini, Gemini 2.5 Pro, and Deepseek-R1.

## B.1 CENTIPAWN CALCULATION USING STOCKFISH

We ran Stockfish v17 (path configurable via `stockfish_path`) in UCI mode with the following settings: fixed analysis depth of 20 plies, no time limit per move, a single thread, 128 MB hash size, MultiPV=1, and skill level of 20. We convert the engine's `Cp` or `Mate` score to centipawns via a standardized function: centipawn values directly for `Cp` evaluations, and $\pm 1000$ for any mate score: positive for winning mates, negative for losing mates. Blunder, Mistake, and Inaccuracy thresholds are based on Lichess's Win% cutoffs: 30%, 20%, and 10% respectively (Lichess, 2023). These hyperparameters provide consistent, interpretable per-ply metrics while keeping analysis costs tractable.

## B.2 DRAGON 1 SETTINGS

All Dragon 1 experiments were run on the following computer: Windows 11, WSL 2, Core i5 13600KF, 64GB DDR5 RAM, RTX4090. As we use it, the Dragon chess engine is stochastic: to verify this, we ran 1000 games between Dragon skill 1 vs skill 2. We found that game metrics such as player material count and game duration variate significantly (standard deviation is 10-40% of the mean).

## B.3 MODEL INFORMATION

In Table 4 we map all the API model names and additional settings (e.g., quantization) to their cleaned name used in the paper. Note that all open source models not run through an API (e.g., groq) were run with quantization on a RTX 4090.

In Table 5 we show the average cost per game across models on our leaderboard where cost was tracked, across all games.

## B.4 ELO CALCULATIONS

To calculate Elo, we played at least 33 total games against varying skill levels in Dragon 1 with the following models: o3 (low), Grok 3 Mini (high), o4-mini, o3-mini. We provide Win/Loss and number of games against each skill level in Table 7. Note that we played o3 (low) and Grok 3 Mini (high) against skills 1-5 (each $\geq$ 169 games), o3-mini (high) and o4-mini (high) against skills 1-2 (each $\geq$ 49 games), and the rest of the models against skill 1 (each $\geq$ 33 games). We also played o3 (low) against skill 10 because we found that it performed quite well against skill 5 (71.9% Win/Loss). However, we found that against skill 10, o3 (low) only achieved a 3.0% Win/Loss, meaning even the most powerful model we thoroughly tested still has a ways to go.

Pseudocode for the Elo calculations resulting in the values in Figure 3 is in Algorithm 1, which takes in a list of opponents with their Elo and corresponding win (1), draw (0.5), loss (0) and calculates an estimate for the LLM's Elo and a 95% confidence interval. Notably, when calculating Elo we add a correction of 35 points to correct for the fact that the LLMs always play as black. We base this on analysis finding that white empirically wins about 54% of games when facing an opponent of the same rating, which equates to 35 points[4].

## B.5 ELO COMPARISONS

We base our Elo scores on Dragon 1, which has different skill levels each paired with a chess.com Elo estimate. Due to this, we can compare the LLMs with players on chess.com. Chess world champion Magnus Carlsen has an active profile at chess.com as the player with the highest Elo rating of 2839 (Chess.com, 2025c). Additionally, on average, a chess.com user has an Elo rating of 611.10 based on 63,120,101 total players (Chess.com, 2025b). Of our evaluated models, only o3 (low) is

---

[4]https://en.chessbase.com/post/the-sonas-rating-formula-better-than-elo

---

**Algorithm 1** Estimate True Elo Rating

---

**Require:** Records $R = \{(R_i, S_i)\}_{i=1}^n$         $\triangleright$ $R_i$ opponent Elo, $S_i \in \{0, 0.5, 1\}$
**Require:** White-advantage $W$         $\triangleright$ 35 Elo
**Ensure:** Estimated rating $\hat{R}$ and 95% CI half-width ME
 1: **function** EXPECTEDSCORE$(r, (R_i)_{i=1}^n)$
 2:     **for** $i \leftarrow 1$ **to** $n$ **do**
 3:        $\hat{S}_i \leftarrow 1 / \left(1 + 10^{(R_i - r)/400}\right)$         $\triangleright$ i.e., $E_i(r)$
 4:     **end for**
 5:     **return** $(\hat{S}_i)_{i=1}^n$
 6: **end function**
 7: **function** SCOREDIFF$(r)$
 8:     $\hat{S} \leftarrow$ EXPECTEDSCORE$(r, (R_i)_{i=1}^n)$
 9:     **return** $\sum_{i=1}^n (S_i - \hat{S}_i)$
10: **end function**
11: // 1) Solve for the black rating of the LLM
12: $R_{\text{black}} \leftarrow$ FINDZERO(ScoreDiff, $[\min_i R_i - 400, \max_i R_i + 400]$)     $\triangleright$ find $r$ such that
      ScoreDiff$(r) = 0$ and is within 400 Elo of the min and max opponent Elos
13: // 2) Compute Fisher information at $R_{\text{black}}$
14: $\hat{S} \leftarrow$ EXPECTEDSCORE$(R_{\text{black}}, (R_i)_{i=1}^n)$
15: $\mathcal{I} \leftarrow \sum_{i=1}^n \hat{S}_i (1 - \hat{S}_i) (\ln 10/400)^2$
16: SE $\leftarrow 1/\sqrt{\mathcal{I}}$
17: // 3) Adjust for white-advantage and form 95% CI
18: $\hat{R} \leftarrow R_{\text{black}} + W$
19: ME $\leftarrow 1.96 \times$ SE
20: **return** $(\hat{R}, \text{ME})$

---

able to perform better when compared to the average chess.com player, with all models significantly far away from the upper bound of the top player, showing significant room for improvement.

While human comparison is important, we also include the random agent for additional context, which is used in the first phase of our evaluation. A random agent has an Elo rating of -122.3, calculated when played against 1000 games each of skills 1-4. As expected with their performance against a random agent directly, all players have a higher Elo, though the worst player, o3-mini (low), does not perform much better, as expected. This signifies that the engine can beat the random agent easily and that there are no unexpected effects.

## C    ADDITIONAL RESULTS

### C.1    ABLATIONS

We present full results on all our ablations for Grok 3 Mini (low) and o4-mini (low) in Table 3. We always play 30 games against a random agent with the LLM as black except for the LLM as white setting, where the roles are reversed. We also use the default unicode board in all settings except the No Legal Moves setting. Because the default unicode board does not have all board information (e.g., castling rights), we provide a FEN for No Legal Moves instead, meaning we are comparing to the FEN setting as the No Legal Moves baseline. We also note that each time the LLM fails to select a valid move in `make_move`, it is provided a message with the board state in FEN like `Failed to make move: illegal uci: 'd5e4' in 1k3b2/1p2pp1r/p7/3p4/3r4/8/PKb5/8 b - - 3 35`. So note when we change the board state in our ablations, regardless of what we change it to we still always see this FEN when an illegal move is made.

**Implementation Details**   For Always Board State we remove `get_current_board` from the list of actions and instead always provide the board state in the prompt. For Always Legal Moves we do the same but for `get_legal_moves`. For Only `make_move` we remove both

`get_current_board` and `get_legal_moves` from the list of actions and instead include the board state and legal moves in the prompt, leaving `make_move` as the only action. This mimics a non-agentic scenario since there is only one action needed in every conversation, so each should only have one turn unless a mistake is made in making a move. We present examples of ASCII and FEN (Forsyth–Edwards Notation) boards below:

---

**Example of ASCII board**

```
rnbqkbnr
pppppppp
........
........
.P......
........
P.PPPPPP
RNBQKBNR
```

---

**Example of FEN board**

```
rnbqkbnr/pppppppp/8/8/6P1/8/PPPPPP1P/RNBQKBNR b KQkq - 0 1
```

---

For No Legal Moves, we simply remove `get_legal_moves` and replace the unicode board with a FEN board. For Previous Moves, we include all previous moves in an ordered list in UCI notation before the Game Loop Prompt. Here, it is black's turn and there have been 10 full moves and 21 plys:

---

**Previous Moves Prompt**

Previous moves (UCI): 1. e2e3 g8f6, 2. a2a4 e7e5, 3. e1e2 b8c6, 4. b1a3 f8e7, 5. a3b1 e5e4, 6. b2b3 e8g8, 7. c1a3 d7d5, 8. g2g4 f6g4, 9. a3d6 e7d6, 10. d1e1 g4e5, 11. b1a3

---

For Previous Moves + Only `make_move`, we use the Only `make_move` setting but prepend the Previous Moves Prompt in the same way as for Previous Moves.

**Analysis**   Overall, we see for our Actions ablations, performance always increases for both models when we choose to remove actions and include their information in the prompt instead, suggesting that the models still struggle to choose the actions they need in the agentic system.

For Board Representation, we see Grok 3 Mini (low) performance is robust to changes from unicode to ASCII or FEN, while for o4-mini (low) ASCII is 15% better than unicode and FEN is 6.7% better than ASCII. We also see that when the LLM is the white player performance increases as expected, but still remains below 90% for both models.

When Changing Information, we see removing the ability to query for legal moves decreases performance by almost 30% for Grok 3 Mini (low) and almost 10% for o4-mini (low) compared to the FEN baseline. This shows that o4-mini (low) has a better grasp of the legal moves, but both models struggle, as expected. We see that while including previous moves improves the Win/Loss of both models, it also decreases the average Blunder rate (Table 8). In fact, while o4-mini (low) only improves by 3.4% in Win/Loss over the baseline, there is a large drop in blunders of 9.6%, meaning that including previous moves helps the model avoid larger mistakes during play. When including previous moves in the Only `make_move` setting, we see similar but slightly worse performance than in Only `make_move`, suggesting when the model is only focused on making the next move without needing to call other actions for information, the previous moves either don't help or slightly harm performance.

### C.2   MOA EXPERIMENTS

The Mixture-of-Agents (MoA) approach is defined by a set of proposer (worker) models that are each prompted to provide an answer, then a synthesizer model meant to combine them. For the

latter, there is an aggregator that works by independently querying the list of proposer models and concatenating their outputs into a single message. This context is fed to the synthesizer model, which uses the following system prompt:

---

**MoA Synthesizer System Prompt**

You will be provided with a set of responses from various open-source models to the latest user query.
Your task is to synthesize these responses into a single, high-quality response in British English spelling.
It is crucial to critically evaluate the information provided in these responses, recognizing that some of it may be biased or incorrect.
Your response should not simply replicate the given answers but should offer a refined, accurate, and comprehensive reply to the instruction.
Ensure your response is well-structured, coherent and adheres to the highest standards of accuracy and reliability.

---

In the main experiments we presented MoA results for only o4-mini. However, we now include additional experiments with different ensembles of reasoning and non-reasoning models. We found that none of the tested non-reasoning models when used as both the proposer and synthesizer (Claude Haiku 3.5, GPT-4.1-mini) improved on game proficiency (0 win rate vs random agent) while also improving on instruction following (100% game duration, meaning all of the games completed naturally, i.e. they were not interrupted due to problems like hallucinated moves). We also tried to use o4-mini (low) as the synthesizer instead of o4-mini (medium) as in the main results but it failed, not providing a valid action and instead commenting on the quality of the proposers' responses. Furthermore, we ran experiments using reasoning models with instruction-following issues (Deepseek R1, Gemini 2.5 Pro) among the proposers and a synthesizer strong in instruction-following but weaker in reasoning (GPT-4.1-mini). We found this setup significantly boosted win rates compared to using the reasoning models alone due to recovered instruction following, achieving 100% game duration. Results with the Win/Loss vs a random agent and the game duration are in Table 9.

### C.3 LLMs with 0% Win/Loss

In Table 6, we include all models we ran with 0% Win/Loss (35 models) versus a random opponent that attempted to complete 30 games. We excluded any games with timeout or API errors. For these models, all losses are due to instruction-following failures with models making too many invalid actions or conversation turns.

### C.4 Full Results

For direct comparisons, in the main body we presented results for LLMs vs Random on 30 games. However, to increase the reliability of our evaluation, we ran an increased amount of games on a variety of models. We include results for all games we ran along with the number of games for each result in Table 10. We see that even with more games, the general ranking of models and pattern remains the same: reasoning models perform best, while non-reasoning models struggle to reach over 50% Win/Loss.

### C.5 Comparison with Other Reasoning Benchmarks

Large language models excel on standard reasoning benchmarks: for instance, OpenAI's o1 model achieves 11.1 out of 15 (74%) on the AIME with a single sample per problem, 12.5 out of 15 (83%) using self-consistency over 64 samples, and 13.9 out of 15 (93%) after re-ranking 1000 samples via a learned scorer (OpenAI, 2024). These scores exceed the performance of the majority of AIME participants; for comparison, scoring 10 or above typically places a student in the top 5% of test-takers nationally. On programming contests like Codeforces, o1 attains an Elo of 1258 (62nd percentile) in its preview release and 1673 (89th percentile) in its main version, surpassing most active competitors on the platform.

To compare our performance directly with a real task, we calculate the correlation between our Elo scores versus LiveCodeBench (Jain et al., 2025) performance on the intersection of all models in our chess engine experiments and the LiveCodeBench leaderboard. LiveCodeBench is a popular benchmark for competitive programming where reasoning models perform well. We take the available Pass@1 scores on the benchmark website for comparison. We find that the scores have a Pearson correlation coefficient of 0.686 (p-value: 0.0888), indicating a moderately strong positive correlation between scores on either benchmark. The performance comparison is visualized in Figure 5.

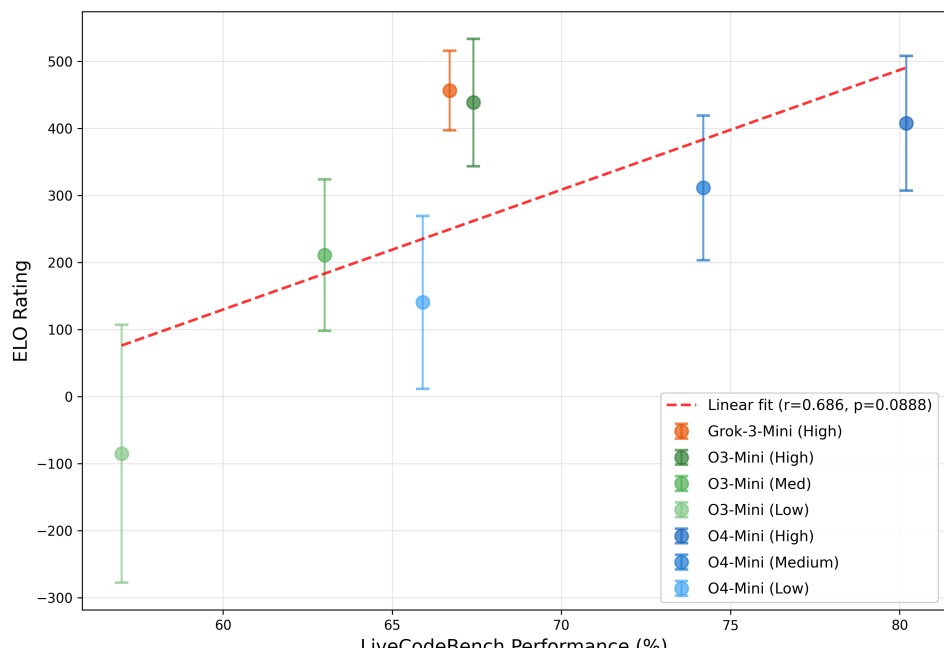

Figure 5: LiveCodeBench Pass@1 scores vs. LLM Chess Elo estimates.

In stark contrast to performance on code and math tasks, when evaluated on our interactive chess benchmark, the LLM we evaluated peaked at Elo 758 against an engine calibrated to chess.com, corresponding to a skill level similar to that of an average online chess player. This contrast underscores a key insight: while LLMs can exceed the abilities of most humans in math and coding competitions, they exhibit a striking weakness in real-time, multi-step strategic environments like chess. Our benchmark surfaces these limitations by requiring not only domain knowledge but also agentic consistency, planning, and game state awareness.

## C.6   ERROR ANALYSIS

During games, we observe various instruction-following issues. These consist primarily of models responding with non-parsable text, where an action can't be identified by simple string matching (i.e., wrong actions), or models requesting illegal moves when issuing a parsable `make_move` action (i.e., wrong moves). Evaluation of conversation traces shows that wrong moves are typically attributed to models' inability to respond with relevant actions, filling the response with verbosity and failing to recognize the desired response format. Wrong moves can be attributed to hallucinations; e.g., even with prior `get_legal_moves` requests and a list of available legal moves in the context, the model can still fail to request a legal move, choosing one not allowed or not listed in the previous message instead.

All games interrupted due to issues can be categorized as one of the following:

1. **Too many wrong actions**: The model produced more than two responses that the game bot failed to parse or make a valid move

2. **Max turns reached**: While deciding on a next move, the chat completions dialog lasted for more than 10 turns. This typically indicates repetitive loops, such as going in circles with actions like `get_current_board` and `get_legal_moves`.

3. **Model Errors**: These consist of errors such as timeouts when a model failed to respond within a reasonable amount of time or when a specific API code was returned. Connectivity and infrastructure issues are discarded (log deleted) and the corresponding games are rerun.

On a subset of our benchmark with 76 evaluated models, 54 out of 76 (71.1%) models experienced abnormal finishes. Table 11 shows the average breakdown of failure reasons, and Table 12 presents the average mistake rates per 1000 moves. The primary driver of failure is making too many wrong actions, responsible for 64.79% of the failures. Per move, wrong actions occur 62.1% of the time as opposed to wrong moves occurring 37.9% of the time. These results indicate that most failures are from models unable to call the correct tools rather than making illegal moves or getting stuck in a repetitive interaction loop.

## D  IMPLEMENTATION DETAILS

Here we include all prompts supplied to the model, as well as a sample dialog for a single move. Below is the prompt that initiates the conversation with the LLM:

> **Game Loop Prompt**
>
> You are a professional chess player and you play as black. Now is your turn to make a move. Before making a move you can pick one of the following actions:
> - `get_current_board` to get the schema and current status of the board
> - `get_legal_moves` to get a UCI formatted list of available moves
> - `make_move <UCI formatted move>` when you are ready to complete your turn (e.g., `make_move e2e4`)
> Respond with the action.

Reflection prompt for when the chess environment can't parse an action in the LLM response:

> **Reflection Prompt for Unparseable Action**
>
> Invalid action. Pick one, reply exactly with the name and space delimitted argument: `get_current_board`, `get_legal_moves`, `make_move <UCI formatted move>`

Reflection prompt for illegal move request by LLM:

> **Reflection Prompt for Illegal Move**
>
> Failed to make move: illegal uci: 'b6c5' in
> `r1bqk2r/pppp1ppp/1N2n1P1/4pKb1/8/P4Q1N/1PPPP1R1/R1B2B2 b kq -`
> `12 25`

Sample dialog of a conversation the LLM has in the agentic environment to output a single move (the board is encoded in Unicode characters and provided to the LLM):

> **Sample Dialog for Single Move**
>
> ```
> Proxy (to Player_Black):
> You are a professional chess player and you play as black.
> Now is your turn to make a move. Before making a move you can
> ```

```
pick one of the following actions:
- 'get_current_board' to get the schema and current status of
the board
- 'get_legal_moves' to get a UCI formatted list of available
moves
- 'make_move <UCI formatted move>' when you are ready to
complete your turn (e.g., 'make_move e2e4')
Respond with the action.

Player_Black (to Proxy):
get_current_board

Proxy (to Player_Black):
♖♘♗♕♔♗♘♖
♙♙♙♙♙♙♙♙
O O O O O O O O
O O O O O O O O
O O O O O O O O
O O O O O O O O
♟♟♟♟♟♟♟♟
♜♞♝♛♚♝♞♜

Player_Black (to Proxy):
get_legal_moves

Proxy (to Player_Black):
a7a6, b7b6, c7c6, d7d6, e7e6, f7f6, g7g6, h7h6, a7a5, b7b5,
c7c5, d7d5, e7e5, f7f5, g7g5, h7h5

Player_Black (to Proxy):
make_move e7e5

Proxy (to Player_Black):
Move made, switching player
```

## E  OPENAI REASONING MODEL TIMEOUTS

OpenAI reasoning models exhibited occasional timeout errors at higher levels of reasoning effort.
They were the only models we tested that often failed to return a response within the default AG2
client timeout of 10 minutes, throwing the following error:

```
TimeoutError: OpenAI API call timed out. This could be due to
congestion or too small a timeout value. The timeout can be
specified by setting the 'timeout' value (in seconds) in the
llm_config (if you are using agents) or the OpenAIWrapper
constructor (if you are using the OpenAIWrapper directly).
```

In all cases, no retries were made. For random opponents these games were excluded, but against
Dragon 1 they were treated as losses for the LLM. As we focus on real-world chess performance,
it is reasonable to enforce consistent time limits and thus assigning a loss should a player fail to
make a move. We note that these issues are likely due to OpenAI's server or the way it handles high
reasoning efforts. Timeout issues are the reason for the lower ranking of some OpenAI reasoning
models when tested with higher reasoning efforts.

Increasing the timeout did not solve the issue. We suspect that some of the game prompts triggered failure modes in models, just like some games states and corresponding prompts provoked hallucinated moves in non-reasoning models.

The the statistics on timeout errors observed while testing Dragon 1 vs o3-mini, o3, and o4-mini are in Table 13.

Table 4: API name and settings (e.g., quantization, reasoning effort) mapped to the clean model name used in the paper. If quantized, we ran locally.

| API Name and Settings | Cleaned Model Name |
|---|---|
| gpt-4-0613 | GPT-4 |
| qwen2.5-7b-instruct-1m | Qwen2.5-7B-Instruct |
| internlm3-8b-instruct | InternLM3-8B-Instruct |
| qwen-max-2025-01-25 | Qwen2.5-Max |
| qwen2.5-14b-instruct@q8_0 | Qwen2.5-14B-Instruct (Q8) |
| qwq-32b | QWQ-32B |
| o3-2025-04-16-low | o3 (low) |
| gpt-4o-2024-08-06 | GPT-4o (2024-08-06) |
| mistral-nemo-12b-instruct-2407 | Mistral-Nemo-Instruct-2407 |
| gpt-35-turbo-1106 | GPT-3.5 Turbo (11/06) |
| o1-preview-2024-09-12 | o1-preview |
| grok-3-mini-beta-high | Grok 3 Mini (high) |
| claude-v3-5-sonnet-v1 | Claude 3.5 Sonnet |
| amazon.nova-lite-v1 | Amazon Nova Lite |
| gemini-2.0-flash-exp | Gemini 2.0 Flash (exp) |
| o4-mini-2025-04-16-low | o4-mini (low) |
| llama-3-70b-instruct-awq | Llama-3-70B-Instruct |
| gpt-4.5-preview-2025-02-27 | GPT-4.5 |
| deepseek-chat-v3 | DeepSeek-V3 |
| gemma-2-27b-it@q6_k_l | Gemma 2 27B |
| llama3.1-8b | Llama-3.1-8B |
| claude-v3-5-haiku | Claude 3.5 Haiku |
| qwen2.5-72b-instruct | Qwen2.5-72B-Instruct |
| gpt-4.1-nano-2025-04-14 | GPT-4.1 Nano |
| granite-3.1-8b-instruct | Granite-3.1-8B-Instruct |
| llama3-8b-8192 | Llama-3-8B |
| gemma2-9b-it-groq | Gemma 2 9B |
| qwen-turbo-2024-11-01 | Qwen Turbo |
| gpt-4o-2024-11-20 | GPT-4o (2024-11-20) |
| amazon.nova-pro-v1 | Amazon Nova Pro |
| o1-2024-12-17-low | o1 (low) |
| qwen-plus-2025-01-25 | Qwen Plus |
| gpt-35-turbo-0301 | GPT-3.5 Turbo (03/01) |
| mercury-coder-small | Mercury Coder Small |
| deephermes-3-llama-3-8b-preview@q8 | DeepHermes-3-Llama-3-8B-Preview |
| o4-mini-2025-04-16-high | o4-mini (high) |
| gpt-4o-mini-2024-07-18 | GPT-4o Mini |
| gpt-4-turbo-2024-04-09 | GPT-4 Turbo |
| o4-mini-2025-04-16-medium | o4-mini (medium) |
| gemini-2.5-pro-preview-03-25 | Gemini 2.5 Pro Preview |
| gpt-4-32k-0613 | GPT-4 32K |
| phi-4 | Phi-4 |
| gemini-2.0-flash-thinking-exp-1219 | Gemini 2.0 Flash Thinking |
| mistral-small-instruct-2409 | Mistral-Small-Instruct-2409 |
| mistral-small-24b-instruct-2501@q4_k_m | Mistral-Small-24B-Instruct-2501 |
| llama-2-7b-chat | Llama-2-7B-Chat |
| gemma-3-12b-it@iq4_xs | Gemma 3 12B (iq4) |
| claude-v3-7-sonnet-thinking_10000 | Claude 3.7 Sonnet Thinking |
| gemini-1.5-flash-001 | Gemini 1.5 Flash |
| deepseek-chat-v3-0324 | DeepSeek-V3 (0324) |
| deepseek-reasoner-r1 | Deepseek-R1 |
| llama-4-scout-cerebras | Llama 4 Scout |
| chat-bison-32k@002 | Chat-Bison-32K |
| qwen2.5-14b-instruct-1m | Qwen2.5-14B-Instruct |
| o1-2024-12-17-medium | o1 (medium) |
| claude-v3-haiku | Claude 3 Haiku |
| grok-3-mini-beta-low | Grok-3 Mini (low) |
| o3-mini-2025-01-31-low | o3-mini (low) |
| llama-3.1-tulu-3-8b@q8_0 | Llama-3.1-Tulu-3-8B |
| gpt-4o-2024-05-13 | GPT-4o (2024-05-13) |
| gpt-35-turbo-0125 | GPT-3.5 Turbo (01/25) |
| claude-v3-7-sonnet | Claude 3.7 Sonnet |
| gemma-2-9b-it-8bit | Gemma 2 9B (8bit) |
| gpt-35-turbo-0613 | GPT-3.5 Turbo (06/13) |
| gemini-2.0-flash-lite-preview-02-05 | Gemini 2.0 Flash Lite (preview) |
| o3-mini-2025-01-31-medium | o3-mini (medium) |
| gpt-4.1-2025-04-14 | GPT-4.1 |
| gemini-2.0-flash-lite-001 | Gemini 2.0 Flash Lite |
| o3-2025-04-16-medium | o3 (medium) |
| gemini-2.0-flash-001 | Gemini 2.0 Flash |
| deepseek-r1-distill-qwen-14b@q8_0 | DeepSeek-R1-Distill-Qwen-14B |
| ministral-8b-instruct-2410 | Mistral 8B Instruct |
| deepseek-r1-distill-qwen-32b@q4_k_m | DeepSeek-R1-Distill-Qwen-32B |
| llama-3.3-70b | Llama-3.3-70B |
| grok-2-1212 | Grok-2 |
| gemma-3-12b-it@q8_0 | Gemma 3 12B (q8) |
| gemma-3-27b-it@iq4_xs | Gemma 3 27B |
| claude-v3-5-sonnet-v2 | Claude 3.5 Sonnet v2 |
| gpt-4.1-mini-2025-04-14 | GPT-4.1 Mini |

Table 5: Average tokens per move and cost per game for models where cost was tracked. Note that some models are excluded, e.g., when run locally or token counting was handled differently. Note that some costs are lower due to poor performance and resulting early termination.

| Model | Avg. Tokens/Move | Avg. Cost/Game |
|---|---|---|
| o3 (low) | 1927.5 | $8.1653 |
| o4-mini (high) | 5695.2 | $2.7146 |
| o3 (medium) | 5040.3 | $7.3626 |
| o1 (medium) | 3309.1 | $19.5655 |
| o4-mini (medium) | 2155.6 | $1.1091 |
| o1 (low) | 1638.9 | $13.4843 |
| o4-mini (low) | 680.23 | $0.5273 |
| o3-mini (medium) | 2337.8 | $1.6058 |
| o1-preview | 2660.1 | $22.5618 |
| Claude 3.7 Sonnet Thinking | 671.33 | $2.0754 |
| Claude 3.7 Sonnet | 262.81 | $0.8993 |
| GPT-4 32K | 6.66 | $2.2266 |
| Claude 3.5 Sonnet v2 | 91.15 | $0.5590 |
| Qwen2.5-Max | 6.06 | $0.1336 |
| GPT-4 Turbo | 6.06 | $0.8482 |
| GPT-4o (2024-11-20) | 51.59 | $0.3165 |
| GPT-4.1 | 18.94 | $0.1976 |
| GPT-4.5 | 8.03 | $6.4834 |
| GPT-4 | 8.21 | $1.8986 |
| Claude 3.5 Haiku | 67.72 | $0.0465 |
| GPT-4o (2024-08-06) | 7.7 | $0.2081 |
| Claude 3.5 Sonnet | 88.13 | $0.5658 |
| Gemini 2.5 Pro Preview | 434.93 | $0.5570 |
| GPT-4o (2024-05-13) | 31.34 | $0.2669 |
| o3-mini (low) | 669.8 | $0.4827 |
| Deepseek-R1 | 4585 | $0.9375 |
| GPT-4.1 Mini | 8.2 | $0.0172 |
| GPT-4o Mini | 104.64 | $0.0215 |
| Llama-3-70B-Instruct | 41.61 | $0.0205 |
| Gemini 2.0 Flash | 93.77 | $0.0147 |
| Grok-2 | 66.23 | $0.1904 |
| Gemini 1.5 Flash | 19.91 | $0.0034 |
| Gemma 2 27B | 55.04 | $0.0199 |
| Gemma 2 9B (8bit) | 58.12 | $0.0014 |
| DeepSeek-V3 (0324) | 410.71 | $0.0470 |
| Llama-3.3-70B | 102.98 | $0.0140 |
| Qwen Plus | 440.41 | $0.0728 |
| Qwen2.5-72B-Instruct | 219.47 | $0.0110 |
| Gemini 2.0 Flash (exp) | 168.15 | $0.0115 |
| Llama-3.1-8B | 162.1 | $0.0009 |
| Gemini 2.0 Flash Lite | 150.15 | $0.0075 |
| DeepSeek-V3 | 246.93 | $0.0258 |
| Amazon Nova Lite | 534.38 | $0.0000 |
| Amazon Nova Pro | 177.19 | $0.0000 |
| Chat-Bison-32K | 31.64 | $0.0000 |
| Claude 3 Haiku | 210.64 | $0.0000 |
| DeepHermes-3-Llama-3-8B-Preview | 101.36 | $0.0014 |
| DeepSeek-R1-Distill-Qwen-14B | 3073.1 | $0.0019 |
| DeepSeek-R1-Distill-Qwen-32B | 2173.8 | $0.0020 |
| Gemini 2.0 Flash Lite (preview) | 144 | $0.0044 |
| Gemini 2.0 Flash Thinking | 724.54 | $0.0010 |
| Gemma 2 9B | 20.22 | $0.0020 |
| Gemma 3 12B (iq4) | 111.14 | $0.0000 |
| Gemma 3 12B (q8) | 151.11 | $0.0000 |
| Gemma 3 27B | 115.84 | $0.0000 |
| GPT-3.5 Turbo (01/25) | 77.01 | $0.0020 |
| GPT-3.5 Turbo (03/01) | 67.06 | $0.0012 |
| GPT-3.5 Turbo (06/13) | 93.63 | $0.0027 |
| GPT-3.5 Turbo (11/06) | 48.32 | $0.0011 |
| GPT-4.1 Nano | 31.51 | $0.0010 |
| Granite-3.1-8B-Instruct | 469.13 | $0.0029 |
| InternLM3-8B-Instruct | 1543.9 | $0.0125 |
| Llama-2-7B-Chat | 116.31 | $0.0001 |
| Llama-3.1-Tulu-3-8B | 1996.3 | $0.0013 |
| Llama-3-8B | 57.02 | $0.0004 |
| Mercury Coder Small | 837.84 | $0.0327 |
| Mistral 8B Instruct | 72.11 | $0.0000 |
| Mistral-Nemo-Instruct-2407 | 47.7 | $0.0000 |
| Mistral-Small-24B-Instruct-2501 | 110.95 | $0.0000 |
| Mistral-Small-Instruct-2409 | 88.24 | $0.0003 |
| Phi-4 | 333.54 | $0.0006 |
| Qwen Turbo | 192.37 | $0.0016 |
| Qwen2.5-14B-Instruct | 235.27 | $0.0085 |
| Qwen2.5-14B-Instruct (Q8) | 150.63 | $0.0096 |
| Qwen2.5-7B-Instruct | 140.79 | $0.0001 |
| QWQ-32B | 8158 | $0.0433 |

Table 6: LLMs with a 0% Win/Loss on 30 games along with the reasons for their losses. Note that none of these models were able to complete games but instead always lost due to instruction-following failures. Reasoning models are shaded.

| Model | Too Many Wrong Actions | Max Turns |
|---|---|---|
| Amazon Nova Lite | 76.7 | 23.3 |
| Amazon Nova Pro | 100.0 | 0.0 |
| Claude 3 Haiku | 10.0 | 90.0 |
| Chat-Bison-32K | 100.0 | 0.0 |
| DeepHermes-3-Llama-3-8B-Preview | 96.7 | 3.3 |
| DeepSeek-R1-Distill-Qwen-14B | 100.0 | 0.0 |
| DeepSeek-R1-Distill-Qwen-32B | 73.3 | 26.7 |
| Gemini 2.0 Flash Lite (preview) | 100.0 | 0.0 |
| Gemini 2.0 Flash Thinking | 100.0 | 0.0 |
| Gemma 2 9B | 100.0 | 0.0 |
| Gemma 3 12B (iq4) | 100.0 | 0.0 |
| Gemma 3 12B (q8) | 100.0 | 0.0 |
| Gemma 3 27B | 100.0 | 0.0 |
| GPT-3.5 Turbo (01/25) | 100.0 | 0.0 |
| GPT-3.5 Turbo (03/01) | 100.0 | 0.0 |
| GPT-3.5 Turbo (06/13) | 100.0 | 0.0 |
| GPT-3.5 Turbo (11/06) | 100.0 | 0.0 |
| GPT-4.1 Nano | 100.0 | 0.0 |
| Granite-3.1-8B-Instruct | 60.0 | 40.0 |
| InternLM3-8B-Instruct | 60.0 | 40.0 |
| Llama-2-7B-Chat | 100.0 | 0.0 |
| Llama-3.1-Tulu-3-8B | 23.3 | 76.7 |
| Llama-3-8B | 90.0 | 10.0 |
| Llama-3.1-8B | 80.0 | 20.0 |
| Mercury Coder Small | 100.0 | 0.0 |
| Mistral 8B Instruct | 100.0 | 0.0 |
| Mistral-Nemo-Instruct-2407 | 100.0 | 0.0 |
| Mistral-Small-24B-Instruct-2501 | 100.0 | 0.0 |
| Mistral-Small-Instruct-2409 | 100.0 | 0.0 |
| Phi-4 | 100.0 | 0.0 |
| Qwen Turbo | 100.0 | 0.0 |
| Qwen2.5-14B-Instruct | 70.0 | 30.0 |
| Qwen2.5-14B-Instruct (Q8) | 96.7 | 3.3 |
| Qwen2.5-7B-Instruct | 100.0 | 0.0 |
| QWQ-32B | 93.3 | 6.7 |

Table 7: Total number of games played against each skill along with Win/Loss for all games playing against that skill.

| Model | Skill | Total Games | Win/Loss |
|-------|-------|-------------|----------|
| o3 (low) | 1 | 33 | 81.8 |
| | 2 | 33 | 72.7 |
| | 3 | 33 | 75.8 |
| | 4 | 33 | 68.2 |
| | 5 | 32 | 71.9 |
| | 10 | 33 | 3.0 |
| Grok 3 Mini (high) | 1 | 33 | 51.5 |
| | 2 | 34 | 48.5 |
| | 3 | 34 | 41.2 |
| | 4 | 34 | 38.2 |
| | 5 | 34 | 25.0 |
| o4-mini (high) | 1 | 27 | 61.1 |
| | 2 | 22 | 56.8 |
| o3-mini (high) | 1 | 31 | 67.7 |
| | 2 | 26 | 57.7 |
| o4-mini (medium) | 1 | 40 | 53.8 |
| o3-mini (medium) | 1 | 38 | 39.5 |
| o4-mini (low) | 1 | 33 | 30.3 |
| o3-mini (low) | 1 | 33 | 10.6 |

Table 8: Average Blunder rate (%) per ply when including previous moves vs baseline. Lower is better.

| Model | LLM CHESS | Previous Moves |
|-------|-----------|----------------|
| Grok 3 Mini (low) | 9.1 | **3.5** |
| o4-mini (low) | 11.2 | **1.6** |

Table 9: Performance of different MoA configurations on game playing tasks. Win/Loss shows the win rate against a random agent, and Game Duration shows the percentage of games that completed naturally without interruption. We run with the following configurations: 1) Deepseek R1 MoA. Workers: Deepseek-R1, GPT-4.1-mini (temp 0.3), GPT-4.1-mini (temp 1.0); Synthesizer: GPT-4.1 (temp 0.3), and 2) Gemini 2.5 Pro MoA. Workers: Gemini 2.5 Pro (preview version, 03-25), GPT-4.1-mini (temp 0.3), GPT-4.1-mini (temp 0.0); Synthesizer: GPT-4.1 (temp 0.3).

| Model | Win/Loss | Game Duration |
|-------|----------|---------------|
| Deepseek R1 | 32.3% | 62.4% |
| Deepseek R1 MoA | **62.9%** | **100%** |
| Gemini 2.5 Pro | 41.9% | 73.6% |
| Gemini 2.5 Pro MoA | **78.9%** | **100%** |

Table 10: Full results for LLM vs. Random on variable number of $\geq 30$ games. Reasoning models are shaded. The percentage of games ending due to checkmate from either side, instruction-following failures, and draws are also displayed.

| Player | Total Games | Win/Loss | Checkmate | Instruction | | Draws | | | |
|---|---|---|---|---|---|---|---|---|---|
| | | | Checkmate | Wrong Actions | Max Turns | Stalemate | Insuff. Material | 5x Repetition | Max Moves |
| o3 (medium) | 48 | 100.0 | 100.0 | 0.0 | 0.0 | 0.0 | 0.0 | 0.0 | 0.0 |
| o3 (low) | 41 | 96.3 | 92.7 | 0.0 | 0.0 | 0.0 | 0.0 | 2.4 | 4.9 |
| o4-mini (high) | 38 | 96.1 | 92.1 | 0.0 | 0.0 | 5.3 | 2.6 | 0.0 | 0.0 |
| o1 (medium) | 40 | 91.2 | 82.5 | 0.0 | 0.0 | 10.0 | 2.5 | 0.0 | 5.0 |
| Grok 3 Mini (high) | 44 | 86.4 | 72.7 | 0.0 | 0.0 | 4.5 | 4.5 | 0.0 | 18.2 |
| o4-mini (medium) | 159 | 84.3 | 68.6 | 0.0 | 0.0 | 11.9 | 12.6 | 0.0 | 6.9 |
| o1 (low) | 47 | 78.7 | 57.4 | 0.0 | 0.0 | 6.4 | 19.1 | 0.0 | 17.0 |
| o4-mini (low) | 74 | 70.9 | 44.6 | 0.0 | 0.0 | 17.6 | 9.5 | 0.0 | 28.4 |
| o1-preview | 30 | 68.3 | 46.7 | 10.0 | 0.0 | 3.3 | 20.0 | 0.0 | 20.0 |
| o3-mini (medium) | 44 | 67.0 | 36.4 | 2.3 | 0.0 | 20.5 | 4.5 | 0.0 | 36.4 |
| Claude 3.7 Sonnet Thinking | 37 | 62.2 | 24.3 | 0.0 | 0.0 | 0.0 | 18.9 | 0.0 | 56.8 |
| Grok 3 Mini (low) | 52 | 58.7 | 21.2 | 0.0 | 0.0 | 13.5 | 1.9 | 0.0 | 63.5 |
| Gemini 2.5 Pro Preview | 33 | 53.0 | 36.4 | 27.3 | 3.0 | 15.2 | 9.1 | 0.0 | 9.1 |
| GPT-4 32K | 33 | 48.5 | 3.0 | 0.0 | 0.0 | 0.0 | 0.0 | 0.0 | 97.0 |
| Qwen2.5-Max | 60 | 48.3 | 3.3 | 0.0 | 0.0 | 0.0 | 0.0 | 0.0 | 96.7 |
| GPT-4o (2024-11-20) | 71 | 47.9 | 12.7 | 0.0 | 0.0 | 0.0 | 0.0 | 0.0 | 87.3 |
| Claude 3.5 Sonnet v2 | 60 | 47.5 | 8.3 | 3.3 | 0.0 | 1.7 | 0.0 | 0.0 | 86.7 |
| Claude 3.5 Sonnet | 60 | 46.7 | 18.3 | 1.7 | 0.0 | 0.0 | 0.0 | 0.0 | 80.0 |
| GPT-4 Turbo | 30 | 46.7 | 6.7 | 0.0 | 0.0 | 0.0 | 0.0 | 0.0 | 93.3 |
| GPT-4.5 | 44 | 46.6 | 6.8 | 0.0 | 0.0 | 0.0 | 0.0 | 2.3 | 90.9 |
| GPT-4 | 33 | 45.5 | 9.1 | 0.0 | 0.0 | 0.0 | 0.0 | 0.0 | 90.9 |
| GPT-4o (2024-08-06) | 59 | 44.1 | 15.3 | 0.0 | 0.0 | 1.7 | 0.0 | 0.0 | 83.1 |
| GPT-4.1 | 80 | 43.8 | 13.8 | 1.2 | 0.0 | 0.0 | 0.0 | 0.0 | 85.0 |
| Claude 3.5 Haiku | 42 | 42.9 | 7.1 | 2.4 | 4.8 | 2.4 | 0.0 | 0.0 | 83.3 |
| Claude 3.7 Sonnet | 42 | 40.5 | 16.7 | 11.9 | 0.0 | 2.4 | 0.0 | 0.0 | 69.0 |
| GPT-4o (2024-05-13) | 60 | 40.0 | 11.7 | 8.3 | 0.0 | 0.0 | 0.0 | 0.0 | 80.0 |
| o3-mini (low) | 56 | 37.5 | 7.1 | 19.6 | 8.9 | 3.6 | 0.0 | 0.0 | 60.7 |
| Deepseek-R1 | 31 | 32.3 | 22.6 | 51.6 | 6.5 | 3.2 | 9.7 | 0.0 | 6.5 |
| GPT-4.1 Mini | 84 | 30.4 | 9.5 | 3.6 | 26.2 | 0.0 | 0.0 | 0.0 | 60.7 |
| GPT-4o Mini | 30 | 30.0 | 3.3 | 36.7 | 0.0 | 0.0 | 0.0 | 0.0 | 60.0 |
| Llama-3-70B-Instruct | 30 | 25.0 | 3.3 | 46.7 | 0.0 | 0.0 | 0.0 | 0.0 | 50.0 |
| Gemini 2.0 Flash | 67 | 21.6 | 10.4 | 55.2 | 0.0 | 0.0 | 0.0 | 0.0 | 34.3 |
| Grok-2 | 49 | 19.4 | 6.1 | 63.3 | 0.0 | 0.0 | 0.0 | 0.0 | 30.6 |
| Gemini 1.5 Flash | 30 | 16.7 | 6.7 | 60.0 | 0.0 | 0.0 | 0.0 | 0.0 | 33.3 |
| Gemma 2 27B | 30 | 13.3 | 6.7 | 66.7 | 0.0 | 0.0 | 0.0 | 0.0 | 26.7 |
| Llama 4 Scout | 39 | 10.3 | 2.6 | 64.1 | 12.8 | 0.0 | 0.0 | 0.0 | 20.5 |
| Gemma 2 9B (8bit) | 30 | 6.7 | 3.3 | 83.3 | 0.0 | 0.0 | 0.0 | 0.0 | 13.3 |
| DeepSeek-V3 (0324) | 45 | 5.6 | 2.2 | 88.9 | 2.2 | 0.0 | 0.0 | 0.0 | 6.7 |
| Llama-3.3-70B | 42 | 4.8 | 9.5 | 73.8 | 7.1 | 0.0 | 0.0 | 0.0 | 9.5 |
| Qwen Plus | 33 | 4.5 | 0.0 | 90.9 | 0.0 | 0.0 | 0.0 | 0.0 | 9.1 |
| Gemini 2.0 Flash (exp) | 30 | 3.3 | 0.0 | 90.0 | 3.3 | 0.0 | 0.0 | 0.0 | 6.7 |
| Qwen2.5-72B-Instruct | 30 | 3.3 | 3.3 | 90.0 | 0.0 | 0.0 | 0.0 | 0.0 | 6.7 |
| Gemini 2.0 Flash Lite | 66 | 1.5 | 4.5 | 95.5 | 0.0 | 0.0 | 0.0 | 0.0 | 0.0 |
| DeepSeek-V3 | 70 | 1.4 | 1.4 | 90.0 | 5.7 | 0.0 | 0.0 | 0.0 | 2.9 |

Table 11: ==Average breakdown of failure reasons across abnormal finishes.==

| Failure Reason | Percentage |
|---|---|
| Too many wrong actions | 64.79% |
| Max turns reached | 13.96% |
| Error | 21.25% |

Table 12: ==Average mistake rates per 1000 moves.==

| Mistake Type | Per 1000 Moves | Percentage |
|---|---|---|
| Wrong actions | 122.70 | 62.1% |
| Wrong moves | 74.86 | 37.9% |

Table 13: Number of timeout errors in OpenAI reasoning models when facing Dragon 1 opponents with varying skill levels. The default timeout is 10 minutes.

| Opponent Skill Level | LLM | Total logs | Errors |
|---|---|---|---|
| 1 | o3 (low) | 33 | 0 |
| 1 | o3-mini (low) | 33 | 0 |
| 1 | o3-mini (medium) | 38 | 0 |
| 1 | o3-mini (high) | 33 | 2 |
| 1 | o4-mini (low) | 33 | 0 |
| 1 | o4-mini (medium) | 40 | 0 |
| 1 | o4-mini (high) | 33 | 6 |
| 2 | o3 (low) | 33 | 0 |
| 2 | o3-mini (high) | 30 | 4 |
| 2 | o4-mini (high) | 30 | 8 |
| 2 | o4-mini (high) w/ 20m timeout | 29 | 7 |
| 2 | o4-mini (high) w/ 60m timeout | 6 | 4 |
| 3 | o3 (low) | 33 | 0 |
| 4 | o3 (low) | 33 | 0 |
| 5 | o3 (low) | 35 | 0 |
| 10 | o3 (low) | 33 | 0 |
| 10 | o3 (medium) w/ 60m timeout | 11 | 2 |

