# OpenReview forum: "LLM CHESS: Benchmarking Reasoning and Instruction-Following in LLMs through Chess"
_ICLR.cc/2026/Conference — Submitted to ICLR 2026_

### Official Review · Reviewer_cZAL · 2025-10-29

**Soundness:** 3
**Presentation:** 2
**Contribution:** 2
**Rating:** 4
**Confidence:** 3

**Summary:**

LLM CHESS introduces an agentic, chess-based benchmark to jointly evaluate reasoning and instruction-following in 50+ LLMs using full-game play, per-ply quality metrics, and engine-grounded Elo, with code and a public leaderboard released for reproducibility. Empirically, most models struggle even versus a random opponent while reasoning-enhanced models fare better.

**Strengths:**

1. The choice of chess as the testbed is conceptually solid. It naturally embodies combinatorial search, long-horizon planning, and rule-based reasoning, making it a meaningful domain.
2. The analyses and experiments are extensive. The consistent advantage of reasoning-enhanced “thinking” models over standard LLMs provides credible support for the benchmark’s claims.
3. The framework is reproducible and extensible, with open code, public leaderboards, and adjustable opponent strengths, allowing the benchmark to evolve as models improve.

**Weaknesses:**

1. Most LLMs obtain nearly zero Win/Loss in Table 4, suggesting that the current difficulty curve may be poorly calibrated. It remains unclear whether the benchmark measures reasoning limitations or simply overwhelms models with excessive interaction complexity.
2. Figure 1 conveys little information, with too much large white space and minimal data illustration. Core analyses such as the ablation in experiments should be added into the main passage instead of in the appendix.
3. The agentic interface itself adds heavy cognitive and formatting burdens. Since removing it in ablations leads to more than 20 % performance gains, failures may be caused by API-understanding errors rather than reasoning problems.
4. The paper implicitly equates chess reasoning with general reasoning ability, yet with no cross-task validation (e.g., MATH or BBH scores). The validity of reasoning in this benchmark remains unverified.
5. Writing quality and narrative flow are weak.

**Questions:**

1. The evaluation focuses exclusively on LLMs; including non-LLM or rule-based baselines would anchor what the result actually represents and clarify how the benchmark scales across architectures.
2. Can this framework extended to other board games?
3. Runtime and API costs are not reported, which is necessary to assess practicality and reproducibility.
4. Because the benchmark uses open interaction protocols, will targeted fine-tuning on its trajectories quickly inflate leaderboard scores, without improving underlying reasoning?

---

> ### Author Response · Authors · 2025-11-22
>
> Thank you for your thoughtful feedback. We've outlined responses to your concerns below:
>
> ---
>
> ### `W1 & W3: Entanglement of reasoning ability with interaction complexity`
>
> The goal of our benchmark is to evaluate both instruction-following and reasoning abilities in chess. We acknowledge that these two concepts may be entangled. However, we believe the design of the benchmark is still valuable towards drawing conclusions regarding these abilities. Importantly, by designing a simple agentic setting requiring consistent tool calling and valid moves (described in Section 2.1), we focus on a realistic setting where both instruction-following and reasoning are necessary. Our core belief driving these choices is that a model with very good reasoning ability but poor instruction-following will seldom be used.
>
> We believe that our two phases, of LLMs vs random player (Section 3.1) then LLMs vs chess engine (Section 3.2), helps us to isolate instruction-following and reasoning, respectively. We consider the first phase less of a full reasoning evaluation and more of a sanity check to ensure that models are strong enough at instruction-following to behave reliably in our agentic setting. We argue that beating a random player is something we should expect most current LLMs to do on a regular basis given their stated reasoning abilities. The fact that many LLMs we evaluate can’t even reach a 50% Win/Loss signifies that there are pressing problems with how the LLMs behave (Figure 2). This shows that these LLMs have not even reached the bare minimum of what we would expect in a simple agentic setting in chess. While we might not be able to evaluate pure reasoning ability in the models that perform poorly in phase 1, we can still make conclusions about reaching the sufficient level of instruction-following that we require.
>
> Then, for the second phase, we generally focus on models that were only able to complete the first phase without instruction-following errors, so can better isolate reasoning performance. For example, in Table 8 we see that o3, o4-mini, and Grok 3 Mini all have 0% instruction-following failures over all games vs random players. We note that o3-mini (low) and o3-mini (medium) still have instruction-following errors, so perhaps these models are thus underestimated in their performance. However, these models generally follow the expected rankings when we scale their reasoning effort, meaning reasoning is having a similar effect as it does in o4-mini. We note the other models are strong enough to pass the base level of instruction-following that we require. Thus we can say that reasoning makes up most performance differentials in the second phase, not instruction-following.
>
> We have updated Section 2.1 to be clearer about the differences.
>
> ---
>
>
> ### `W2: Lack of information in Figure 1 and ablation studies in appendix`
>
> Thank you for pointing these out. Regarding the ablation studies, we have moved the corresponding table to the main body.  For Figure 1, we will enhance it to show more detail in a future version.
>
> ---
>
> ### `Q1: Lack of including non-LLM baselines`
>
> We agree that including non-LLM baselines would help to increase explainability in our benchmark. We offer the following baseline performance levels into our leaderboard for comparison:
> - Chess world champion Magnus Carlsen has an active profile at chess.com and has Elo rating of 2839
> - The average chess.com user has an Elo rating of 618
> - Random player vs Dragon has a rating of -122.3 (when played against 1000 games each of skills 1-4)
>
> We note that of our evaluated models, only o3 (low) is able to perform better when compared to the average chess.com player, with all models thus showing significant room for improvement. We include these baselines in the updated version of our paper and adjust the analysis accordingly in Section 3.2 and Appendix B.5.
>
> ---
>
> ### `Q2: Can you extend this benchmark to other board games`
>
> We believe that this framework can be extended to other board games, as long as there is an engine or way to control the difficulty of the opposing non-LLM player. We believe future work could do this and thus give more power to claims of generalization.
>
> ---
>
> ### `Q3: No report of runtime and API costs`
>
> To keep track of costs, we have created a user-friendly leaderboard website that reports an average game cost. For example, a single game simulation for o3 (low) cost $8.16.
>
> For background, each game log contains accounting of prompt and completion tokens that we used to calculate costs in the leaderboard. Though the logs contain response timings, we found that such times could vary significantly as often we used free API tiers and so saw more variation.
>
> We include a summary of the cost information displayed in the leaderboard in Appendix B.3 in the updated version.
>
> ---

---

> ### Author Response · Authors · 2025-11-22
>
> ---
>
> ### `Q4: Will targeted fine-tuning quickly inflate leaderboard scores?`
>
> As with many tasks, we anticipate that fine-tuning would increase performance as the model learns the interaction setting and more knowledge about chess. However, we expect after a certain point, this improvement will stop. A key difference between our chess benchmark and other benchmarks is that chess is much more immune to memorization given its dynamic nature and combinatorial properties. Because of this, the games are dynamic: it is realistic for new board states to be seen during inference that were not in the training data. Training on every board state is infeasible in chess, as opposed to knowledge-based benchmarks that could all potentially be memorized. Hence we speculate that after some point of fine-tuning, increased general reasoning ability would be necessary to improve performance.
>
> ---
>
> ### `W4: Implicitly equates chess reasoning with general reasoning`
>
> We acknowledge that we have emphasized the importance of reasoning in chess, and have noted that in some sense we can test the generalization of reasoning with this benchmark. By this we do not claim that good performance in chess equates to good general reasoning abilities, but instead that if we claim to have a general reasoner (which may be attributed to the reasoning-enhanced LLMs), it should be able to perform sufficiently well at chess. We also believe chess has insights that may be of general interest and be understood well by the public, as AI models historically have frequently used chess as a signal of progress.
>
> Though we don’t run any experiments on other tasks outside of chess, we emphasize that the performance trends we witness are similar to those in popular math or coding benchmarks, though significantly less impressive. We see that non-reasoning models in general perform poorly against even a player making random moves, with no model reaching 50% Win/Loss in Figure 2. However, powerful reasoning models (e.g., o3) are able to beat the random player almost everytime. These large differentials correspond to the large differences observed in reasoning vs non-reasoning models on other benchmarks. Additionally, in Figure 4a, we see that scaling the reasoning effort results in increased performance, as expected.
>
> To compare our performance directly with a real task, we calculate the correlation between our Elo scores versus LiveCodeBench [1] performance on the intersection of all models in our chess engine experiments and the LiveCodeBench leaderboard. LiveCodeBench is a popular benchmark for competitive programming where reasoning models perform well. We take the Pass@1 score for comparison. We find that the scores have a pearson correlation coefficient of 0.686 (p-value: 0.0888), indicating a moderately strong positive correlation between scores on either benchmark. The corresponding table is below. Results are presented and discussed in Section 3.3 and Appendix C.5 in the updated version.
>
> | Model              | LiveCodeBench Score | ELO Rating |
> |--------------------|--------------------:|-----------:|
> | Grok-3-Mini (High) | 66.7               | 456.35     |
> | o3-Mini (High)     | 67.4               | 438.52     |
> | o3-Mini (Med)      | 63.0               | 210.75     |
> | o3-Mini (Low)      | 57.0               | -85.30     |
> | o4-Mini (High)     | 80.2               | 407.61     |
> | o4-Mini (Medium)   | 74.2               | 311.11     |
> | o4-Mini (Low)      | 65.9               | 140.31     |
>
>
> Together, these results suggest that our chess benchmark reflects the general trends on other reasoning tasks, albeit in their current form LLMs aren’t able to reach world-class performance in chess.
>
> ---
>
> ### References
>
> [1] Jain, N., Han, K., Gu, A., Li, W. D., Yan, F., Zhang, T., ... & Stoica, I. (2025). LiveCodeBench: Holistic and contamination free evaluation of large language models for code. In International Conference on Learning Representations (ICLR 2025).

---

> > ### Comment · Reviewer_cZAL · 2025-11-27
> >
> > I think the authors’ response has addressed most of my concerns. I initially believed that general reasoning ability required stronger validation, but the authors have convinced me that this work meaningfully complements existing reasoning benchmarks and reveals capabilities that current evaluations fail to test. Therefore, I will raise my score.

---

### Official Review · Reviewer_y5WN · 2025-10-30

**Soundness:** 3
**Presentation:** 3
**Contribution:** 2
**Rating:** 6
**Confidence:** 4

**Summary:**

This paper benchmarks language models as chess-playing agents, finding that reasoning models perform at near the median player on chess.com (looking briefly online at the distribution of elo scores.) Most of the models without reasoning (long think chains) fail to beat an agent playing random moves.

I think there are fair concerns about why/whether we would use an LLM for a task like this, but it seems like this paper offers a resource that should exist and I think that people will enjoy seeing.

**Strengths:**

* The paper is clear and easy to follow
* Overall feels like a solid work
* I like how you setup Figure 4a (and b)

**Weaknesses:**

* The primary contribution is the resource here. However, it is unclear what information/inferences use of the resource will offer. What will future users of the benchmark learn from the results?
(See questions)

**Questions:**

* Q: What lessons would you draw from this work? What lessons would you draw from future results on this benchmark?
* Chess was/is a good test bed for reasoning, but now we have models that are pretty good at reasoning, and we have chess models that are good at chess. It is not clear why or where this benchmark fits in. There is an analogy to arithmetic but I'm not sure it fully answers the question.
* Q: Have you seen this `https://www.kaggle.com/game-arena`? Any comments or comparisons?

* I would recommend using something other than "Random Player". It sounds like you're randomly picking human players off of chess.com or something. Perhaps "Random Agent"

---

> ### Author Response · Authors · 2025-11-22
>
> Thank you for your valuable insights and feedback. We address your concerns below:
>
> ---
>
> ### `W1 and Q1: Lacks detail on what we can learn from the benchmark`
>
> The main goal of the benchmark is to evaluate LLMs on reasoning and instruction-following using chess as a testbed. We believe this is important for testing if reasoning abilities in LLMs, largely tested on math and code, can generalize to the domain of chess, which has an important history in testing AI. After testing a variety of models, we find that 1) instruction-following performance varies greatly among models in our agentic setting, signifying that even in a simple agentic setting, reasoning performance may be constrained by the ability to follow instructions, 2) reasoning models perform better, making much less instruction-following errors and seeing increased performance with scaling reasoning effort, and 3) unlike on current math and coding benchmarks, top models show relatively low performance on LLM Chess.
>
> As an example, o4-mini (high) achieves a score of 92.7 on AIME 2025, an elite high school math competition benchmark [1] but achieves only a 407.61 Elo on LLM Chess. While o4-mini is able to perform better than most of the top high school math competitors, on LLM Chess it reaches that of only a beginner player, below the average chess.com player who has an Elo of 618. For context, Magnus Carlsen is the highest ranked player with an Elo of 2839, and the highest player under 20 has Elo of 2768 as of November 19 [2]. This illustrates a stark contrast: o4-mini (high) is able to achieve world-class performance in math but only reaches that of a beginner in chess. Therefore, our work highlights a limitation of current LLMs to embody general reasoning capabilities, with top performance limited to certain domains. As models get better, we will continue to monitor performance, seeing if smaller models stop having basic instruction-following errors and if the most powerful models can reach world-class performance in chess like they have in other domains. Additionally, given this current low performance and the dynamic nature of chess, we believe that it will be harder to saturate LLM Chess, so it can have more longevity.
>
> We have added comparisons with human Elo scores in Appendix B.5 and correlation with a strong reasoning benchmark (LiveCodeBench) in Appendix C.5 to address where LLM Chess fits in the real world.
>
> ---
>
>
> ### `Q2: Unclear where benchmark fits among reasoning and chess models`
>
> We draw from the history of using chess as a benchmark to evaluate AI models. The main idea is not that we believe all reasoning models should be trained to be good at chess. However, for a model to be a general reasoner, we believe it should have sufficiently strong performance across many domains, one of which is chess. Currently, from the information we have, it doesn’t seem that LLM reasoning models are being trained on games like chess, with much of a focus on coding and math. We believe at this point in time, this fact allows us to measure one aspect of the generalizability of reasoning abilities (though noting that good performance at chess does not imply good general reasoning performance). We believe that chess can be used as another signal, perhaps more grounded in the real-world and understandable to the general public, to help keep track of the progress LLMs are making.
>
> ---
>
> ### `Q3: Comparison to existing game-arena benchmark`
>
> The attached game-arena benchmark aligns with our view that chess can be useful as a measure of general model ability. The primary difference between their benchmark and ours is that they use direct LLM-to-LLM comparison, where we focus on LLMs playing against the same non-LLM (either an agent taking random moves or a chess engine). This gives us a more fixed comparison. Additionally, we model our LLMs as agents that can utilize tools, which allows us to evaluate more instruction-following capabilities. We have included this in the related work in the updated version.
>
> ---
>
> Thank you for the suggestion about the “Random Player” label; we have adjusted it to be "Random Agent" in the updated version.
>
> ---
>
> ### References
>
> [1] OpenAI. "Introducing OpenAI o3 and o4-mini." OpenAI, 16 Apr. 2025
>
> [2] Live Chess Ratings & Chess Rankings (November 2025)." Chess.com

---

### Official Review · Reviewer_RA9T · 2025-11-05

**Soundness:** 3
**Presentation:** 3
**Contribution:** 3
**Rating:** 4
**Confidence:** 4

**Summary:**

This study proposes LLM CHESS, a novel and comprehensive benchmark framework designed to evaluate the reasoning and instruction-following capabilities of Large Language Models (LLMs) within the complex, strategic domain of chess. Its core methodology utilizes an agentic interaction setup where LLMs play full games via tool calls: `get_current_board`, `get_legal_moves`, and `make_move`. The study presents a large-scale evaluation of over 50 models. It first assesses their baseline capabilities and instruction adherence by playing against a random agent. Subsequently, high-performing models are pitted against a chess engine (Komodo's Dragon 1) with configurable skill levels to estimate their Elo ratings. Key findings reveal a significant performance gap between so-called "reasoning-enhanced" LLMs and standard ones. Most models struggle to reliably defeat even a random player due to instruction-following failures. Even the top-performing model achieves an Elo of only ~758, highlighting a stark discrepancy between LLM capabilities in dynamic, strategic environments and their performance on other reasoning tasks like math and programming. To foster future research, the authors have open-sourced the experimental framework, a public leaderboard, and the dataset of games.

**Strengths:**

1. The agentic framework is a key innovation. It not only evaluates the quality of moves but also tests the model's integrated ability to follow instructions and use tools. The design is highly scalable—difficulty can be increased by simply raising the opponent's skill level—ensuring the benchmark's long-term relevance.
2. The breadth of the study, covering over 50 models, is commendable. This large scale provides a robust foundation for drawing conclusions about the current state of LLMs on this task, making the findings more convincing.
3. The well-designed ablation studies offer deep insights into why models fail, demonstrating that model performance is highly sensitive to prompting and interaction formats. This points to a lack of robust generalization capabilities.

**Weaknesses:**

1. A core part of the analysis distinguishes between "reasoning-enhanced" and "standard" models. The argument would be strengthened if the paper provided a more explicit and operational definition for this classification (e.g., based on specific test-time algorithms, architectural features, or training methods).
2. The analysis could be enriched by more granular case studies of errors. For instance, analyzing the types of mistakes (distinguishing between simple tactical blunders and deeper strategic misunderstandings) or identifying common patterns in instruction-following failures could offer a more nuanced understanding of the models' cognitive limitations.
3. The default agentic setup concurrently tests strategic thinking, tool use, and format adherence. While the ablation studies help to deconstruct this, it raises the question of whether the benchmark could be designed to isolate these skills more directly. For instance, a model might be a strong strategist but a poor tool-user, and the current primary metrics may not clearly differentiate between these two cases.

**Questions:**

1. Could you provide a more formal or operational definition for "reasoning-enhanced" models as used in this study? Is this distinction based on specific test-time algorithms (e.g., search, self-consistency), architectural features, or particular training methodologies?
2. The high rate of instruction-following failures is a key finding. It would be very informative to see a more fine-grained classification of these errors. For example, do models get stuck in loops (e.g., repeatedly calling get_current_board), fail to parse the available actions, or generate syntactically invalid moves in the make_move tool call? This could help differentiate between failures in attention, comprehension, or action generation.
3. The Mixture-of-Agents (MoA) experiments yield interesting but limited gains, and performance appears highly sensitive to the choice of proposer and aggregator models. Could you please elaborate on the model's sensitivity to this configuration? For instance, what might the results be if a stronger model were used as the aggregator? Furthermore, what specific prompt or mechanism was used by the aggregator to synthesize the proposals?
4. Regarding line 404, does the phrase "by removing actions and instead supplying the removed information automatically" mean that the current board state and legal moves are provided to the LLM automatically, bypassing the need for it to call the corresponding tools?
5. On pages 20 and 21, some of the prompt text extends beyond the page margins. We suggest adjusting the formatting to improve readability.

---

> ### Author Response · Authors · 2025-11-22
>
> Thank you for your valuable insights and feedback. We address your concerns below:
>
> ---
>
> ### `W1 & Q1: Lack of distinction between “reasoning-enhanced” and “standard models`
>
> We define “reasoning-enhanced” models as those that are specifically advertised/characterized by their developers as “reasoning” (e.g. OpenAI) or “thinking” (e.g. Anthropic, Google) without going into detail as to how those models are built (generally RL and test-time compute are mentioned, yet the detail and disclosure varies). On the surface the reasoning enhanced models manifest their nature by splitting the response into two sections: 1) reasoning/thinking intermediary, delimited via special markdown (such as a think tag) or residing in a separate API response section (e.g. thinking block in Anthropic API), and 2) the final answer.
>
> E.g., aligned with their advertised functionalities, we designate as reasoning the following: all “o” family of models (e.g., o1, o3, o4-mini), Claude 3.7 Thinking, Grok 3 Mini, Gemini 2.5 Pro, and Deepseek-R1.
>
> We have included this more detailed description in Appendix B of the updated version.
>
> ---
>
>
> ### `W2 & Q2: Should include more granular case studies of errors`
>
> During games, we find various instruction-following issues such as models responding with non-parsable text, where an action can’t be identified by simple string matching (i.e., wrong actions), or requesting illegal moves by issuing a parsable make_move action (i.e., wrong moves). Evaluation of conversation traces shows that wrong moves are typically attributed to models’ inability to respond with relevant actions, filling the response with verbosity and failing to recognize the desired response format. Wrong moves can be attributed to hallucinations; e.g., even with prior get_legal_moves requests and a list of available legal moves in the context, the model can still fail to request a legal move, choosing one not allowed/not listed in the previous message instead.
>
> All games interrupted due to issues can be categorized as the following:
> - Too many wrong actions: the model produced more than 2 responses that the game bot failed to parse OR make a valid move
> - Max turns reached: while deciding on a next move, the chat completions dialog lasted for more than 10 turns. This typically indicates repetitive loops, such as going in circles with actions like get_current_board/get_legal_moves
> - Model Errors: e.g. timeouts when a model failed to respond within a reasonable amount of time or when a specific API code was returned that means model error. Connectivity and infrastructure issues are discarded (log deleted) and the corresponding games are rerun.
>
> Please see the below for results on a subset of 76 evaluated models. Note there were 54 out of 76 (71.1%) models with abnormal finishes:
>
> Average breakdown of failure reasons (sums to ~100%):
> - **Too many wrong actions:** 64.79%
> - **Max turns reached:** 13.96%
> - **Unknown issue:** 0.00%
> - **Error:** 21.25%
>
> Average mistakes per 1000 moves:
> - **Wrong actions:** 122.70 (62.1%)
> - **Wrong moves:** 74.86 (37.9%)
>
> We have placed this analysis in the updated version in Appendix C.6 and will release our code used to do this analysis upon acceptance.
>
> ---

---

> ### Author Response · Authors · 2025-11-22
>
> ### `W3: Entanglement of instruction-following and reasoning abilities in the benchmark`
>
> The goal of our benchmark is to evaluate both instruction-following and reasoning abilities in chess. We acknowledge that these two concepts may be entangled. However, we believe the design of the benchmark is still valuable towards drawing conclusions regarding these abilities. Importantly, by designing a simple agentic setting requiring consistent tool calling and valid moves (described in Section 2.1), we focus on a realistic setting where both instruction-following and reasoning are necessary. Our core belief driving these choices is that a model with very good reasoning ability but poor instruction-following will seldom be used.
>
> We believe that our two phases, of LLMs vs random player (Section 3.1) then LLMs vs chess engine (Section 3.2), helps us to isolate instruction-following and reasoning, respectively. We consider the first phase less of a full reasoning evaluation and more of a sanity check to ensure that models are strong enough at instruction-following to behave reliably in our agentic setting. We argue that beating a random player is something we should expect most current LLMs to do on a regular basis given their stated reasoning abilities. The fact that many LLMs we evaluate can’t even reach a 50% Win/Loss signifies that there are pressing problems with how the LLMs behave (Figure 2). This shows that these LLMs have not even reached the bare minimum of what we would expect in a simple agentic setting in chess. While we might not be able to evaluate pure reasoning ability in the models that perform poorly in phase 1, we can still make conclusions about reaching the sufficient level of instruction-following and reasoning that we require.
>
> Then, for the second phase, we generally focus on models that were only able to complete the first phase without instruction-following errors, so can better isolate reasoning performance. For example, in Table 8 we see that o3, o4-mini, and Grok 3 Mini all have 0% instruction-following failures over all games vs random players. We note that o3-mini (low) and o3-mini (medium) still have instruction-following errors, so perhaps these models are thus underestimated in their performance. However, these models generally follow the expected rankings when we scale their reasoning effort, meaning reasoning is having a similar effect as it does in o4-mini. We note the other models are strong enough to pass the base level of instruction-following that we require. Thus we can say that reasoning makes up most performance differentials, not instruction-following.
>
> We have updated Section 2.1 to be clearer about the differences.
>
> ---
>
> ### `Q3: Lacks details on MoA experiments`
>
> We provide additional details on our MoA experiments below, which we will place in an updated version of the paper in the near future:
>
> The aggregator works by independently querying a list of proposer/worker models and concatenating their outputs into a single message. This context is fed to the synthesizer model, which operates under a specific system prompt:
>
> ```
> You will be provided with a set of responses from various open-source models to the latest user query.
> Your task is to synthesize these responses into a single, high-quality response in British English spelling.
> It is crucial to critically evaluate the information provided in these responses, recognizing that some of it may be biased or incorrect.
> Your response should not simply replicate the given answers but should offer a refined, accurate, and comprehensive reply to the instruction.
> Ensure your response is well-structured, coherent and adheres to the highest standards of accuracy and reliability.
> ```

---

> ### Author Response · Authors · 2025-11-22
>
> While in the paper we presented MoA results for only o4-mini, we also ran additional experiments with different ensembles of reasoning and non-reasoning models. We found that none of the tested non-reasoning models when used as both proposer and synthesizer (Claude Haiku 3.5, gpt-4.1-mini) improved on game proficiency (0 win rate vs random player) while also improving on instruction following (100% game duration, meaning all of the games completed naturally, i.e. they were not interrupted due to e.g. hallucinated moves). We found it surprising that using o4-mini (low) instead of o4-mini (medium) as in the main experiments as the synthesizer failed to improve win rates. Additionally, using reasoning models with instruction-following issues (Deepseek R1, Gemini 2.5 Pro) as one of the proposers and weak synthesizer (gpt-4.1-mini) did significantly boost win rates due to recovered instruction following, achieving 100% game duration. Results with the Win/Loss vs a random player and the game duration are below and are in Appendix C.2 in the updated version:
>
> | Model | Win/Loss | Game Duration |
> |-------|---------------|---------------|
> | Deepseek R1 | 32.3% | 62.4% |
> | Deepseek R1 MoA | 62.9% | 100% |
> | Gemini 2.5 Pro | 41.9% | 73.6% |
> | Gemini 2.5 Pro MoA | 78.9% | 100% |
>
> **Deepseek R1 MoA Configuration:**
> Worker models: Deepseek-R1, gpt-4.1-mini (temp 0.3), gpt-4.1-mini (temp 1.0). Synthesizer: gpt-4.1 (temp 0.3).
>
> **Gemini 2.5 Pro MoA Configuration:**
> Worker models: Gemini 2.5 Pro (preview version, 03-25), gpt-4.1-mini (temp 0.3), gpt-4.1-mini (temp 0.0). Synthesizer: gpt-4.1 (temp 0.3).
>
> ---
>
>
> ### `Q4 & Q5: Additional clarifications`
>
> Yes, in line 404 we mean that board state and legal moves were provided to the LLM automatically when removed as tools so that the model retains the same amount of information when making its moves. This corresponds to the “Only make_move” setting in Table 6.
>
> Thank you for pointing out the page margin issues on pages 20 and 21, we have fixed this in the updated version.
>
> ---

---

### Author Response · Authors · 2025-11-26
**Summary**

### TL;DR of Our Work
We introduce LLM CHESS, a benchmark designed to evaluate reasoning and instruction-following capabilities of LLMs through chess. We evaluated over 50 models using a simple agentic framework where LLMs play full games. Our evaluation is split into two phases: first using the full set of models against a random agent to assess baseline instruction-following, then using a more targeted set of powerful reasoning models against a chess engine to estimate Elo ratings. Key findings reveal that most models struggle to defeat even a player making random moves, and even the top-performing model we tested achieved only 758 Elo, only slightly higher than an average human player.

In response to the feedback from the reviewers, we have made corresponding improvements in the paper, with the revisions highlighted in yellow.

---

### Reviewers' Recognition

**Reviewers find LLM CHESS well-constructed and valuable.**

- `cZAL`: "The choice of chess as the testbed is conceptually solid. It naturally embodies combinatorial search, long-horizon planning, and rule-based reasoning"
- `RA9T`: "The agentic framework is a key innovation. It not only evaluates the quality of moves but also tests the model's integrated ability to follow instructions and use tools."
- `y5WN`: "The paper is clear and easy to follow”; “Overall feels like a solid work"

**Reviewers appreciate the scale and reproducibility of the evaluation.**

- `RA9T`: "The breadth of the study, covering over 50 models, is commendable. This large scale provides a robust foundation for drawing conclusions"
- `cZAL`: "The framework is reproducible and extensible, with open code, public leaderboards, and adjustable opponent strengths"

**Reviewers find the ablation studies insightful.**

- `RA9T`: "The well-designed ablation studies offer deep insights into why models fail, demonstrating that model performance is highly sensitive to prompting and interaction formats."

---

### Reviewers' Concerns and Our Rebuttals

**`RA9T` requests clearer definition of "reasoning-enhanced" models — addressed.**

- "The argument would be strengthened if the paper provided a more explicit and operational definition for this classification."

We define "reasoning-enhanced" models as those specifically advertised by developers as "reasoning" (e.g., OpenAI) or "thinking" (e.g., Anthropic, Google). These models split responses into reasoning/thinking intermediary sections and final answers. Examples include all "o" family models, Claude 3.7 Thinking, Grok 3 Mini, Gemini 2.5 Pro, and Deepseek-R1. We include this detailed description in Appendix B in the new version.

**`RA9T` and `cZAL` raise concerns about entanglement of instruction-following and reasoning — addressed.**

- `RA9T`: "The default agentic setup concurrently tests strategic thinking, tool use, and format adherence... a model might be a strong strategist but a poor tool-user."
- `cZAL`: "The agentic interface itself adds heavy cognitive and formatting burdens. Since removing it in ablations leads to more than 20 % performance gains, failures may be caused by API-understanding errors rather than reasoning problems."

Our two-phase design helps isolate these abilities. Phase 1 (vs player making random moves) serves as a sanity check for instruction-following; Phase 2 focuses on models that completed Phase 1 with minimal to no instruction-following errors to better isolate reasoning. For example, o3, o4-mini, and Grok 3 Mini all have 0% instruction-following failures. The goal of this is to penalize models with good reasoning but poor instruction-following, which in practice will be difficult to use in real agentic workflows. We clarify the purpose of each phase in Section 2.1 in the updated paper.

**`RA9T` requests more granular error analysis — addressed.**

- "It would be very informative to see a more fine-grained classification of these errors."

We categorized failures across 76 models (54 with abnormal finishes): Too many wrong actions (64.79%), Max turns reached (13.96%), and Model errors (21.25%). Per 1000 moves: wrong actions (122.70, 62.1%) vs wrong moves (74.86, 37.9%). Wrong actions stem from non-parsable responses; wrong moves are hallucinations where models request illegal moves despite having legal moves listed in context. This analysis is described in Appendix C.6 in the updated paper.

---

> ### Author Response · Authors · 2025-11-26
>
> **`cZAL` questions benchmark validity without cross-task correlation — addressed.**
>
> - "The paper implicitly equates chess reasoning with general reasoning ability, yet with no cross-task validation"
>
> We calculated correlation between our Elo scores and LiveCodeBench performance, finding a Pearson correlation of 0.686 (p=0.0888), indicating moderately strong positive correlation. We emphasize that good chess performance doesn't equate to general reasoning, but a claimed "general reasoner" should perform sufficiently well at chess. We have updated the paper with this analysis in Section 3.2 and Appendix C.5.
>
> **`y5WN` questions what lessons can be drawn from the benchmark — addressed.**
>
> - "It is unclear what information/inferences use of the resource will offer."
>
> Our benchmark shows: 1) instruction-following varies greatly even in simple agentic settings, 2) reasoning models perform better with scaling reasoning effort in chess, and 3) unlike math/coding benchmarks, top models show low chess performance. For example, o4-mini (high) scores 92.7 on AIME 2025 but only 407.61 Elo, below the average chess.com player (618 Elo). This highlights limitations in LLMs' general reasoning capabilities. The goal of LLM CHESS is a living benchmark that allows people to understand how well LLMs perform at chess in an easily extensible way, with conclusions about reasoning and instruction-following that could potentially have impacts on other agentic or reasoning tasks. We have added comparisons with human Elo scores in Appendix B.5 and correlation with a strong reasoning benchmark (LiveCodeBench) in Appendix C.5 to address where LLM CHESS fits in the real world.
>
> **`cZAL` requests runtime and API costs — addressed.**
>
> - "Runtime and API costs are not reported, which is necessary to assess practicality and reproducibility."
>
> Our leaderboard reports average game costs (e.g., $8.16 per game for o3-low). Game logs contain token accounting used for cost calculations. We include cost summaries in Appendix B.3 in the updated version.
>
> **`cZAL` raises concerns about fine-tuning inflation — addressed.**
>
> - "Will targeted fine-tuning on its trajectories quickly inflate leaderboard scores"
>
> Unlike knowledge-based benchmarks, chess is immune to memorization given its dynamic, combinatorial nature. Training on every board state is infeasible. After initial fine-tuning gains, increased general reasoning ability would be necessary to improve further.

---

### Meta-Review · Area_Chair_W65o · 2026-01-03

**Summary:**

This paper proposes LLM CHESS, a benchmark that evaluates LLMs as chess-playing agents through full-game interaction with tool calls. The framework jointly tests instruction-following, tool use, and strategic reasoning, and is evaluated at scale with performance summarized via win rates and engine-estimated Elo ratings. Reviewers broadly agree that the benchmark is technically sound, the experiments are extensive, and the released code and leaderboard support reproducibility.

Despite these strengths, the paper does not sufficiently meet the bar for acceptance at ICLR due to limitations in contribution clarity, conceptual positioning, and interpretability of results, even after a thorough rebuttal.

A primary concern raised across reviewers is the unclear scope and impact of the benchmark. While the authors frame LLM CHESS as a way to test reasoning and instruction-following in a dynamic environment, multiple reviewers questioned what concrete scientific insights the benchmark enables beyond confirming that most current LLMs perform poorly at agentic chess. The core findings—that non-reasoning models fail even against a random agent and that reasoning-enabled models perform better but remain weak in absolute Elo—are interesting but largely descriptive, and it remains unclear how future results on this benchmark would advance understanding of LLM reasoning or guide model development in a principled way.

Closely related is the concern that the benchmark entangles multiple failure modes—strategic reasoning, tool use, formatting, and instruction-following—making it difficult to attribute poor performance to specific cognitive limitations. While the authors argue that such entanglement reflects realistic agentic usage and introduce a two-phase evaluation (vs. random agent, then vs. engine) to partially disentangle these factors, several reviewers remain unconvinced that the benchmark cleanly isolates reasoning ability. In particular, ablations showing large performance gains when removing the agentic interface suggest that interaction complexity, rather than chess reasoning per se, may dominate outcomes for many models.

Reviewers also expressed skepticism about the broader relevance of chess-based agentic evaluation in the current LLM landscape. Although the authors emphasize chess’s historical role in AI and present correlations with external reasoning benchmarks (e.g., LiveCodeBench), these analyses are limited in scope and statistical strength. As a result, the paper does not fully establish that performance on LLM CHESS provides a uniquely informative or necessary signal about general reasoning capabilities beyond existing benchmarks, nor does it clearly position itself relative to prior game-arena or gameplay-based evaluations.

**Recommendation: Reject.**
While LLM CHESS is a well-engineered and carefully evaluated benchmark, its incremental nature, entanglement of evaluation dimensions, and limited demonstrated insight beyond descriptive performance gaps place it below the acceptance threshold for ICLR at this time.

**Reviewer Concerns:**

As above

**Reviewer Scores:**

I do not think Reviewer RA9T would change the score.

---

### Decision · Program_Chairs · 2026-01-26

Reject